# Variability of light absorption coefficients by different size fractions of suspensions in the southern Baltic Sea

Justyna Meler[1], Dagmara Litwicka[1], Monika Zabłocka[1]

[1]Institute of Oceanology Polish Academy of Sciences, Powstańców Warszawy 55, 81-712 Sopot, Poland

*Correspondence to*: Justyna Meler (jmeler@iopan.pl)

**Abstract.** Measurements of light absorption coefficients by particles suspended in seawater ($a_p(\lambda)$), by phytoplankton ($a_{ph}(\lambda)$) and detritus ($a_d(\lambda)$) were carried out in the southern Baltic Sea for the original seawater samples and the four size fractions: pico-particles (0.2-2 μm), ultra-particles (2-5 μm), nano-particles (5-20 μm) and micro-particles (20-200 μm). Chlorophyll *a* (Chl*a*) and suspended particulate matter (SPM) concentrations were determined. The proportions of particles from the size

classes in the $a_p(443)$, $a_{ph}(443)$ and $a_d(443)$ were determined. Pico and ultra-particles had the largest contribution to the total particles absorption - an average of 38% and 31%. Particles of 5-20 μm accounted for approximately 20% of $a_p(443)$ and $a_{ph}(443)$ and 29% of $a_d(443)$. The contribution of particles > 20 μm averaged 5-10%. In total SPM contribution of micro-particles averaged 17%, nano, ultra, pico-partlices, 29%, 26%, and 27%, respectively. In total Chl*a*, the proportions of pico and ultra-particles averaged 35% each, nano 16% and micro-paritcles 15%. Temporal and spatial variability of particles

contributions in size classes were observed.

      The average chlorophyll-specific and mass-specific light absorption coefficients, i.e. light absorption coefficients normalized to Chl*a* or SPM , were determined for all size fractions. The chlorophyll-specific coefficients $a_p^{(Chla)}(\lambda)$, $a_d^{(Chla)}(\lambda)$ and $a_{ph}^{(Chla)}(\lambda)$, ± standard deviations, do not allow clear separation of the individual fractions. For mass-specific coefficients, $a_p^{(SPM)}(\lambda)$, $a_d^{(SPM)}(\lambda)$ and $a_{ph}^{(SPM)}(\lambda)$, it is possible to distinguish between large particles (20-200 μm) and small and medium

particles (0.2-20μm). These results will allow monitoring of suspended matter in size classes in optically complex waters of southern Baltic Sea.

## 1 Introduction

The biogeochemical and optical properties of coastal waters vary greatly due to high biological productivity, input of terrestrial material and resuspension of benthic matter (D'Sa, Miller and Del Castillo, 2006; Hoepffner and Sathyendranath, 1992).

Additionally, these regions are particularly sensitive to environmental changes. This is due to changes in the populations of phytoplankton and other particles in these ecosystems. Hence, their hydrographic and biogeochemical conditions can change in a short time. Knowledge of the size structure of phytoplankton populations and the effect of mineral and detritus particle sizes on light absorption is essential to better determine the impact of climate change on coastal marine systems (e.g., Le Quéré et al., 2005).

Seawater suspended particle matter (SPM) is an unknown mixture of organic and inorganic compounds, and its composition varies spatially and temporally as a function of various physical (e.g., tides) and biogeochemical (e.g., phytoplankton blooms) factors (D'Sa and Ko, 2008; Eleveld et al., 2014). Variability of the type and source of particles present in the marine environment implies the variability of its absorption properties in time and space. One of the measures of the organic particles suspended in seawater is the concentration of particulate organic carbon (POC). POC, together with dissolved organic carbon (DOC), contributes to the total organic carbon (TOC) biogeochemical cycle. Particulate organic matter (POM) consists of both, living and non-living matter (microalgae, bacteria, fecal pallets and clays (Volkman and Tanoue, 2002)). The living POM plays an essential role in marine food webs and oceanic biogeochemical cycles through countless processes such as photosynthesis, nitrogen fixation or reminalisation (Voss et al., 2013). The chemical composition of non-living POM is dominated by complex carbohydrates that are difficult to break down; therefore, non-living POM is, in general, characterized by bigger biomass than living POM. The size of suspended particles affects the time of their residence in the water column (larger particles are 'heavier' and sink faster), hence the degree of transformation/demineralization they reach before their suspension to the sea bottom. Particle sinking is the main mechanism of carbon flow below the euphotic zone through so called biological carbon pump (Turner et al., 2015).

Accurate linking of measurements and predictions of particle population characteristics with inherent optical properties (IOP) remains a major scientific challenge, especially in optically complex waters (McKee and Cunningham, 2006; Davies et al., 2014), which also includes the Baltic Sea. The absorption properties of particles suspended in the water column vary significantly depending on their type. The light absorption coefficient by suspended phytoplankton organisms, $a_{ph}(\lambda)$, is related to the composition of pigments contained in algal cells and its spectral shape has two characteristic maxima (approximately 400-490 nm and 660-690 nm). For the remaining suspended particles (detritus and mineral particles), the light absorption spectrum, $a_d(\lambda)$, is characterized by a monotonic decline of exponential shape. The relationships between the coefficients $a_{ph}(\lambda)$ and $a_d(\lambda)$ and the concentration of suspended substances present in the water has been the subject of research of many authors (Babin et al., 2003; Woźniak et al., 2011, 2022; Meler et al., 2016 a,b, 2017, 2018; Castagna et al., 2022). The common knowledge is that changes in the spectral shapes of the absorption coefficients follow the changes in size distribution of particles. For phytoplankton, in particular, it has been shown that small phytoplankton exhibit higher absorption coefficients for short blue wavelengths and more pronounced maxima compared to large phytoplankton cells. The larger the phytoplankton cells, the greater the 'flattening' of the absorption spectrum (Morel and Bricaud, 1981; Sathyendranath et al., 1987; Ciotti et al., 2002).

Chlorophyll $a$ (Chl$a$) concentration is a measure of phytoplankton biomass and, together with the size structure of phytoplankton populations, are key ecological indicators in the marine environment (Platt and Sathyendranath, 2008). Changes in these indicators can help detect how the marine ecosystem may respond to natural variability (e.g., innate climate variations) and antropogenic changes (e.g., anthropogenic climate change). The light absorption properties depend on the physical and chemical nature of the water and its constituents and therefore can provide biogeochemically useful information about the composition of the particulate matter suspended in water.

The characteristics of the spectral coefficient $a_{ph}(\lambda)$ can be used to infer the size of phytoplankton, as well as taxonomic information. These techniques are not as sensitive as direct pigment analyzes, especially in optically complex waters with high spatial and temporal variability both in terms of phytoplankton and detritus concentration and composition (Bricaud and Stramski, 1990; Bricaud et al., 2004; Ciotti et al., 2002; Mouw et al., 2017). However, direct measurements of the light absorption coefficient of suspended particles can provide useful information on the size of phytoplankton, which can be used as a basis for methods for estimating the size structure of phytoplankton populations (Bidigare et al., 1989; Moisan et al., 2011; Organelli et al., 2013; Zhang et al., 2015).

The variability of light absorption coefficients by different size fractions of phytoplankton in the natural environment has so far been described only by Ciotti et al. (2002) for the Bering Sea and the Oregon coast. Ciotti et al. (2002) found that more than 80% of $a_{ph}$ variability can be explained by the combined effects of dominant size class and pigment variability and developed a two-component model that relates seawater samples to phytoplankton size structure. The total light absorption in seawater can be obtained from satellite data using the models of Loisel and Stramski (2000) and Loisel and Poteau (2006) by taking the absorption coefficients by phytoplankton and detritus together with dissolved organic matter obtained from analytical decomposition or nonlinear optimization (Ciotti et al., Bricaud, 2006; Bricaud et al., 2012). Further models of absorption depending on size fractions were proposed by Mouw and Yoder (2010) and Devred et al. (2006, 2011). These models use the concentration of Chl*a* calculated from satellite data and allow one to determine the absorption of particles in two size classes: small (<20 μm) and large (> 20 μm) particles.

Other studies reported by various authors concern the modeling or parameterization of light absorption by phytoplankton indirectly, on the basis of Diagnostic Pigment Analysis (DPA) (Vidussi et al., 2001; Uitz et al., 2006, 2008; Brewin et al., 2010, 2012; Hirata et al., 2008, 2011), where three particle size classes are defined: picoplankton, nanoplankton, and microplankton. Various approaches to identify the size structure of phytoplankton populations from satellite data are detailed in the IOCCG report (2014), which describes the possibilities of developing algorithms for remote determination of the contribution of various functional types of phytoplankton (PFT) included in the total population in the waters under study. The PFT concept is used in the study of a number of ecological and biogeochemical problems, especially in model studies. A specific functional type may represent a group of different species related to each other due to certain distinguished features. This approach is of growing interest as it allows for a more thorough study of the role of phytoplankton in global sea and ocean cycles involving the circulation of major chemical elements such as carbon, nitrogen, sulfur and iron, as well as photosynthesis and primary production.

The above mentioned methods of estimating the contribution of phytoplankton size classes do not work for the Baltic Sea, which is a reservoir classified as optically complex (for the DPA method, the results are presented in Meler et al., 2020). The Baltic Sea is a semi-enclosed basin, shallow, characterized by low salinity and a very large inflow of substances from land. This water basin is also characterized by specific optical properties, different from ocean waters (Kowalczuk et al., 2006, 2015; Meler et al. 2016 a, b, 2017, 2018; Woźniak et al., 2011, 2020, 2022). Therefore, most of the algorithms (mathematical formulas) used to interpret remote observation methods of marine and oceanic environments are not applicable to the Baltic

Sea, as they do not take into account the specificity of its waters and hence are subject to large errors. They should be modified or replaced with new ones. For this purpose, *in situ* studies should be carried out, which would allow to directly determine the

light absorption coefficients not only by phytoplankton, but also by all particles and detritus in various size fractions. The research described in this paper is in line with the objectives and guidelines of the Monitoring and Evaluation Strategy of the Helsinki Commission (HELCOM), which aims to ensure the evaluation and monitoring of data that can be used by HELCOM, both for international and national monitoring. The strategy is designed to ensure data production and dissemination by contracting parties of EU Member States. These countries are obliged to comply with several EU directives, such as the Marine

Strategy Framework Directive (MSFD), the Water Framework Directive (WFD), the Habitats and Birds Directives, the EU Strategy for the Baltic Sea Region (EUSBSR) and the EU Integrated Maritime Policy ( HELCOM, 2013). First of all, the Strategy aims to support ecosystem-based maritime spatial planning (MSP) in the Baltic Sea based on ecosystem. It is done by enabling high-quality spatial data and assessment tools for MSP purposes.

The aim of our research was to investigate the variability of the spectra of the light absorption coefficients by different

size fractions of suspended particles ($a_p$) present in the Baltic Sea, in the division into phytoplankton ($a_{ph}$) and detritus ($a_d$). It was carried out for four distinguished size classes: picoplankton (with diameters of 0.2-2 μm), ultraplankton (2-5 μm), nanoplankton (5-20 μm) and microplankton (20-200 μm) according to the division proposed by Sieburth et al. (1978) and Ciotti et al. (2002). The specific objectives were: 1) to determine the contribution of size classes in the total light absorption by particles, phytoplankton and detritus, 2) to determine the average chlorophyll-specific and mass-specific light absorption

coefficients, i.e., light absorption coefficients normalized to the concentration of Chl*a* or the concentration of suspended particulate matter (SPM) ) for distinguished size classes. Our research may be useful in examining whether the use of SPM and Chl*a* data from MERIS or other optical sensors installed on satellites (e.g. OLCI - Ocean and Land Color Instrument) can be used as "high-quality spatial data" and as a HELCOM regional assessment tool. Additionally, our research may be an introduction to the development of remote sensing algorithms for the estimation of the phytoplankton size structure by use of

relationships between spectral features of phytoplankton and its dominant cell sizes (spectral-based approach). The better knowledge of the mechanisms of the POM deposition to the sediments could be helpful in modeling the pace of that process.

## 2 Materials and methods

### 2.1 Description of study area and water samples

Seawater samples were collected in the southern part of the Baltic Sea, from December 2020 to October 2021. The dataset

includes 22 sets of data collected during 3 research cruises on board the R/V *Oceania* (in February, April and September 2021) and 16 sets of data collected during field experiments on the most protruding part of the pier in Sopot (Poland). Figure 1 shows the location of the measurement stations.

The principal factor affecting the variability of the inherent optical properties of Baltic waters in the euphotic zone is the seasonal cycle of biological activity (Sagan, 1991; Olszewski et al., 1992; Kowalczuk et al., 1999, 2005; Meler et al., 2016

a,b). This cycle is governed by physical, biological and chemical processes, which cause the biomass and species composition to vary with time. As a consequence there are three main phytoplankton blooms: a spring bloom of cryophilous diatoms, which then transforms into a bloom of dinoflagellates; a summer bloom of cyanobacteria; and an autumn bloom of thermophilous diatoms (Thamm et al. 2004, Wasmund et al. 2001, Witek and Pliński 1998). The spring blooms can take place from early March to May, the summer ones in July/ August and the autumn ones from September to October (Wasmund et al. 1996,

Thamm et al. 2004, Wasmund and Uhlig 2003). In winter, biological activity is minimal. The maximum runoff of river waters occurs at the turn of April and May, and it often coincides with the spring bloom of phytoplankton initiated by an increase in air and water temperature and more sunlight. River waters carry large amounts of organic dissolved substances (DOM) and nutrients that enhance phytoplankton blooms. The increased amount of phytoplankton in the surface water layer reduces the transparency of the water. In summer there are periodic floods following very heavy rainfall which together with strong winds

can effect in upwelling, which causes cooler water to rise up from the deep layers of the sea. Such events affect the optical properties of waters in the coastal zone and Gulf of Gdańsk - see Olszewski et al. (1992), Kowalczuk (1999) and Sagan (2008). Locations of sampling stations in our study were selected to obtain the greatest possible diversity of waters in optical terms. The research was carried out mainly in the Gulf of Gdańsk, but also in open waters and coastal waters outside the Bay of Gdańsk, as the weather permits. Cyclical measurements on the Sopot pier were made monthly, from December 2020 to May

2021 and in October 2021, and every 7-14 days in the summer period from June to mid-September 2021. The basic characteristics of the water samples were measured (water transparency using the Secchi disk, temperature, and salinity). Sea water transparency measured with the Secchi disc in the Gulf of Gdańsk varied from 2 m to 15 m, at Sopot Pier from 1.5 m to over 6 m (in some cases the disc was visible to the bottom) and in open and coastal waters from 5.5 m to 18.5 m. Temperature and salinity of water varied respectively in the given regions: 0.8 - 17.7 $^0$C and 0.4 - 7.7, 3.1 - 22.3 $^0$C and 7 - 7.8 and 0.2 -

16.1 $^0$C and 7.2 - 8.0. The seawater samples were fractionated according to the size of the suspended particles in the water in a specially designed filtration set through filters/filter meshes with different pore sizes (20 μm, 5 μm, 2 μm). Filtrates of fractions A, B, and C were obtained. The filtration set consisted of two PVC pipes with a diameter of 20 cm and a height of 38 cm and a diameter of 15 cm and a height of 38 cm, between which Teflon square plates 24x24 cm with hollow holes of 15 cm and 13 cm are placed on a metal mesh to allow free flow of water. A nylon mesh (HydroBios, 20 μm and 5 μm) with a size

of 24x24 cm was placed on a plate, secured with a gasket, pressed against the second plate and filtration was carried out.

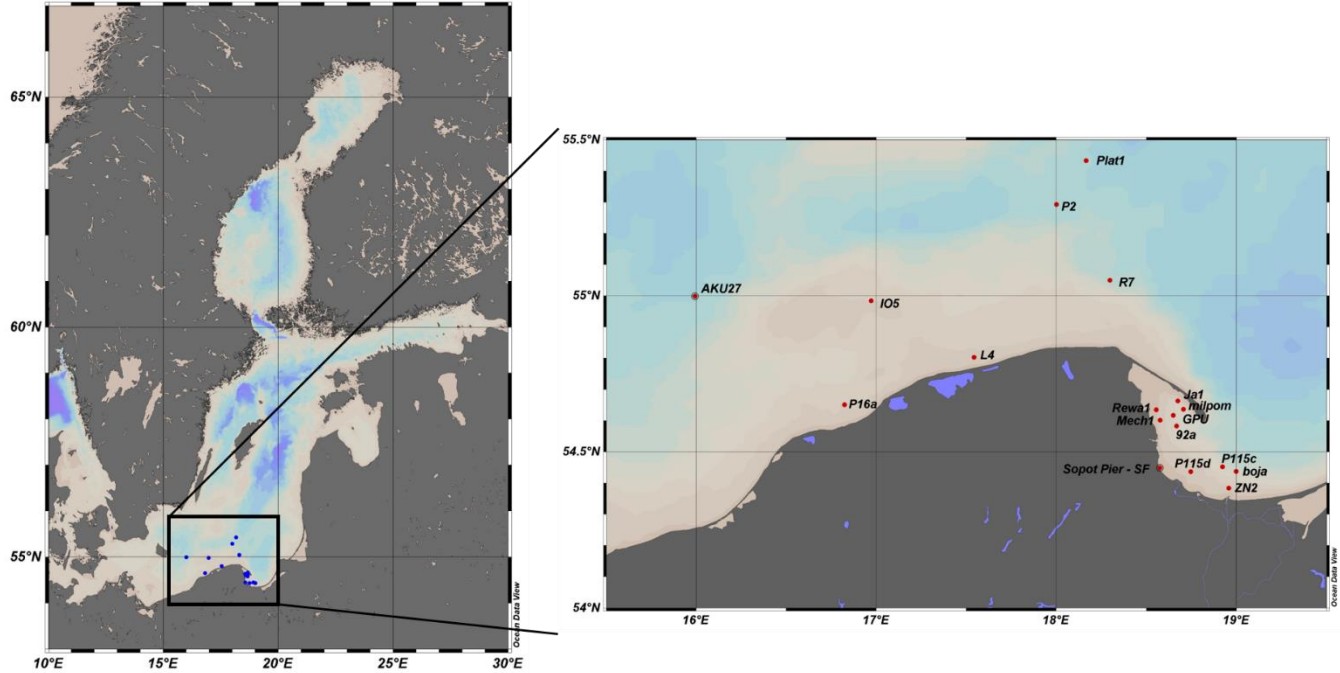

**Figure 1: Location of measurement stations in the Baltic Sea.**

At each location, 40 L to 100 L of surface seawater was collected and integrated into a large tank. 10 L-20 L of water was then extracted and treated as the original (unfiltered) seawater sample. The rest of the water from the integration tank was fractionated by particle size (similar to Koestner et al., 2019). First, the water was gravity filtered through a nylon mesh with 20 μm pores and filtrate of fraction A was obtained. 10 L to 20 L of fraction A was left/cast, and the rest was gravity filtered through a nylon mesh with pores of 5 μm and the filtrate of fraction B was obtained. After separation of the appropriate volume of water for fraction B, the rest of the water was filtered through membrane filters (hydrophilic polycarbonate membrane) Whatmann (Nulepore, 47 mm) or Millipore (Isopore, 47 mm) with pores of 2 μm (under pressure < 0.04 MPa) – the filtrate of fraction C was obtained. Despite the use of specialized nylon meshes with specific mesh sizes, it should be taken into account that filtration of water through them did not guarantee a perfect separation of particle size fractions. Single meshes were used multiple times, after being gently washed with detergent, then rinsed with deionized water, and dried. During the pouring of water, or subsequent washing or rinsing, the meshes could be deformed in any way, causing possible passage of particles with diameters larger than the mesh size to the next filtrate. Moreover, particles suspended in seawater do not have a perfectly spherical shape, this is just an assumption to simplify scientific considerations.

The original seawater samples and fractions A, B and C were filtered to determine the physicochemical parameters: coefficients of light absorption by all suspended particles ($a_p(\lambda)$, m$^{-1}$), detritus ($a_d(\lambda)$, m$^{-1}$) and phytoplankton ($a_{ph}(\lambda)$, m$^{-1}$), concentrations of Chl$a$ (mg m$^{-3}$) and concentrations of SPM (g m$^{-3}$) (including organic (POM) and inorganic (PIM) matter).

The values of individual biogeochemical and optical parameters in all distinguished size classes (micro-size, nano-size, ultra-size and pico-size) were determined as follows. The micro-size was determined by the difference of the original water sample and fraction A. The nano-size was determined by the difference of fraction A and fraction B; the ultra-size was determined from the difference of fraction B and fraction C, while the pico-size was determined from fraction C (the division of phytoplankton particles according to Ciotti et al., 2002).

There have been cases where 'fraction difference' has given a negative value. If the difference between the original water sample and fraction A was negative, it was assumed that the micro-size class was not present hence equal to zero. In this case, fraction A was treated as the value of the original sample. Similar assumptions were used when the differences between fraction A and fraction B or between fraction B and fraction C were negative. It was also assumed that the sum of individual parameters (light absorption coefficients, Chl$a$ and SPM concentrations) obtained for size classes should be equal to the parameters determined for the original, unfractionated seawater sample.

## 2.2 Absorption properties

Spectra of light absorption coefficients by particles suspended in seawater, $a_p(\lambda)$, and by detritus, $a_d(\lambda)$, (particles depigmented by the use of bleach, also called nonalgal particles in the literature), for the original samples and fractions A, B and C were measured using a Perkin-Elmer Lambda 650 spectrophotometer equipped with a 150 mm diameter integrating sphere (measurements inside the sphere). Seawater samples for these analyzes were obtained by filtering small volumes of seawater (from 40 mL for original water samples to 1000 mL for fraction C) through Whatman filters (GF/F, 25 mm). Filters with suspension were stored in liquid nitrogen in a dewar flask and then in a freezer (approx. -20$^0$C) until analyses. Spectrophotometric measurements were performed using a holder in the form of a clip holding a filter with suspension, placed inside the integrating sphere (Stramski et al. 2015, Woźniak et al., 2022). Measurements were performed in the spectral range of 290-860 nm and were carried out in two configurations: for filters with a suspension and for the same filters bleached with a 2% solution of sodium hypochlorite NaClO$_2$ (Meler et al., 2020). Two repetitions were made and averaged. Such spectra were corrected with use of the averaged reference spectra obtained by the measurements on filters through which 30 mL of particle-free seawater was passed. To take into account the lengthening of the optical path of the light falling on the filter with suspension in relation to the same suspension in water, the so-called beta factor was calculated following Stramski et al. (2015). The light absorption coefficients by particles and detritus were determined. From their difference, the light absorption coefficients of phytoplankton ($a_{ph}(\lambda)$) were determined. The precision of the measurement of light absorption coefficients using the IS method in the range of 290 - 860 nm for 3 different filters from the same station was 4.96% ± 2.91%.

The spectra of the light absorption coefficients of the chromophoric dissolved organic matter, $a_{CDOM}(\lambda)$ (m$^{-1}$), were determined spectrophotometrically using a Perkin-Elmer Lambda 650 spectrophotometer. The original seawater samples were first filtered through Whatman filters (GF/F, 47 mm) with nominal pores of 0.7 μm, and then through Sartorius acetone membrane filters (47mm) with pores of 0.2 μm. The filtrate prepared in this way was stored in amber glass bottles in a refrigerator (+4 $^0$C) until further analysis. Measurements were carried out in the spectral range of 200-700 nm in quartz cuvettes

with a 10 cm optical path length. Deionized water was used as a reference. The optical density OD($\lambda$) was converted to CDOM absorption by multiplying OD($\lambda$) by a factor of 2.303 and then dividing by the path length $l$ (m). Finally, the spectra were corrected for residual scattering according to Kowalczuk et al. (2006). The slope coefficients of the light absorption by CDOM in the spectral range of 300-600 nm ($S_{300-600}$, $m^{-1}$) were calculated using the nonlinear least squares fitting that employed the Trust-Region algorithm implemented in Matlab R2013 (Kowalczuk et al., 2006; Stedmon et al., 2000).

## 2.3 Quantities characterizing suspended particulate matter

The concentrations of suspended particulate matter (SPM, g $m^{-3}$), and its organic (POM) and inorganic (PIM) fractions were determined by the gravimetric method and the loss-on-ignition technique (Woźniak et al., 2018; Pearlman et al., 1995). Seawater samples for these analyzes were obtained by filtering seawater (original and fractions A, B, and C) through previously prepared Whatman filters (GF/F, 25 mm). These filters were first combusted at 450 $^0$C for 4.5 h, then rinsed with 0.5 L of pure deionized water and dried for 24 h at 60 $^0$C. After another 24 hours in the desiccator, the filters were weighed and labeled accordingly (Radawag WAX110 microbalance (resolution 0.01 mg). 150 to 2800 mL of seawater was filtered through the filters, depending on the concentration of suspension particles, and then rinsed with 30 mL of clean, deionized water to remove salts from the surface of the suspension on the filter. Filters with suspension were dried at 60 $^0$C for 24 h and stored in a freezer until analysis. In the laboratory, the filters were dried again at 60 $^0$C for 24 h, stored in a desiccator for another 24 h, and then weighed. In this way, the SPM value was obtained. In the next step, the filters were combusted at 450 $^0$C for 4.5 h and weighed. The difference in the weights of the filters with and without the suspension allowed to determination of the SPM concentrations, while the difference in the weights of the filters with the suspension before and after combustion allowed to determination of the POM concentrations. The PIM was calculated from the difference of SPM and POM. The SPM and POM values were also corrected with use of reference filters rinsed with 30 mL of pure deionized water and subjected to the same procedures as the filters with suspension. The measurement precision for 95 % of the triplets was below 15 %, and for all cases the average was 5.83 % ± 4.40 %.

The concentrations of chlorophyll $a$ (Chl$a$, mg $m^{-3}$) were determined by the spectrophotometric method (Lorenzen, 1967). Seawater (original and fractions A, B, and C) was filtered through Whatman filters (GF/F, 47 mm) - from 0.5 L to 7.5 L. The filters were put into liquid nitrogen and then stored in a freezer until analysis. After thawing, the pigments contained in the suspension collected on the filters were extracted in 96% ethanol (8 mL) at room temperature for 24 h (Wintermans and De Mots, 1965; Marker et al., 1980). The extract was centrifuged for 15 minutes and then measurements were made using a Perkin-Elmer Lambda 650 spectrophotometer in 2 cm cuvettes. The 96% ethanol was used as a reference). Chl$a$ concentrations were calculated according to the formula of Stricland and Parsons (1972):

$$Chla(\lambda) = \frac{10^3 \cdot e \cdot (OD(665) - OD(750))}{83 \cdot l \cdot v}, \tag{1}$$

where $e$ is the volume of ethanol ($cm^3$), OD(665) and OD(750) are the optical density at 665 nm and 750 nm respectively, after correction for blank ethanol, 83 (L $g^{-1}$ $cm^{-1}$) is the chlorophyll-specific absorption coefficient for ethanol, $l$ is the length of the

cuvette (cm), $v$ is the volume of filtered seawater (L). Measurement precision for duplicate seawater samples was 5.3 % ± 1.5 %.

## 3 Results

### 3.1 Variability of the biogeochemical properties of suspended matter

The analyzed dataset is characterized by significant variability of biogeochemical parameters defining suspended matter in seawater, both for original water samples and divided into size classes micro-size (20-200 μm), nano-size (5-20 μm), ultra-size (2-5 μm) and pico-size (<2 μm). Table 1 summarizes this variability in detail and presents mean values with standard deviations (SD), and minimum and maximum values. Within 10 months of empirical data collection, a wide variability of SPM concentrations from 0.39 to 15.24 g m$^{-3}$ and Chl$a$ concentrations from 0.18 to 6.85 mg m$^{-3}$ was obtained for the original water samples. Figure 2 shows the average contribution of suspended matter and Chl$a$ concentration in a given size class to total SPM or total Chl$a$ for individual samples. The analyzed data set was divided due to the sampling area: the Gulf of Gdańsk (GG), and extracted from the GG - Sopot Pier (SF) (which shows time variability over 10 months and is the only one that takes into account the summer season), open and coastal waters (OCW) In addition, the data were divided due to the season of sampling. In the case of cruise data, the division is as follows: February - winter, April- spring, September - autumn. In the case of data from Sopot Pier, the data are presented as separate group covering all 4 seasons: winter (December 21 - March 20), spring (March 21 - June 22), summer (June 23 - September 20) and autumn (September 21 - December 20). For the 14 samples (collected during cruises in February and April), it was not possible to separate the ultra and picoplankton fractions, because the amount of suspension of 2-5 μm clogged the membrane filters. For time reasons related to the sampling and filtering of large volumes of seawater for the determination of SPM (3-replicants) and Chl$a$, separation of the ultra and picoplankton fractions was abandoned for these 14 samples. In Figure 2, this 0.2-5 μm fraction is treated as one and marked with a dark gray dashed area.

The average contribution of the SPM in a given size class in the total SPM in the collected data set is similar in the pico-size, ultra-size and nano-size classes of particles - always above 25 %, while the average contribution of micro particles was about 20 % (left panel of Figure 2). It should be noted that in the total SPM in given size fractions (further referred to as micro-, nano-, ultra- and picoplankton for simplicity) there are not only phytoplankton cells, but also detritus and mineral particles. In the first part of Table 2, the contributions of SPM in size classes up to the total SPM for all data and divided into sampling regions are presented. The variability of particles defined as microplankton for all data was 0-59 %, but a contribution in the total SPM greater than 40 % was observed only for 3 cases (for samples collected at the Sopot Pier - 2 cases in summer during phytoplankton bloom and 1 case in early spring). The contribution of nano-particles varied in the range of 0-67 % (contribution > 40% was observed for 9 cases), ultra-particles in the range of 1-50 % (contribution > 40% was observed for 5 cases ), and pico-particles in the range of 5-51 % (contribution > 40% was observed for 7 cases ). In winter, when there is minimal biological activity in the Baltic Sea, it can be seen that the largest contribution in SPM (> 50 %) had particles < 5 μm,

which can be seen in both GG and OCW. In the spring, this trend continued in OCW, while the contribution of micro and nano particles increased in GG. In the autumn period, particles < 5 μm again had the largest contribution in the GG, with a predominance of ultra particles. In the case of OCW, the contribution of small particles decreased and nano and micro particles (> 5 μm) > 60 % dominated. The Sopot Pier dataset shows temporary variability over 10 months and is the only one that takes

into account the summer season. It can be seen that in most cases, in winter, spring and autumn, the contribution of small (< 5 μm) and medium and large (> 5 μm) particles in the SPM is comparable, except for SF06. In summer (SF08-SF12 and SF15) particles < 5 μm contributed the most to SPM, with pico particles predominating, only during phytoplankton blooms, where large algae gathered at the beach, samples were dominated by micro particles, and the proportion of small particles was below 20 %.

**Table 1: Variability of parameters characterizing suspended matter in seawater (mean ± standard deviation (SD), and range of variation)**

| quantity (n=38) | SPM [g m$^{-3}$] | POM [g m$^{-3}$] | PIM [g m$^{-3}$] | Chl$a$ [mg m$^{-3}$] | POM/SPM | Chl$a$/SPM |
|---|---|---|---|---|---|---|
| original samples (all particles, unfiltered) | | | | | | |
| average ± SD | 2.35 ± 2.83 | 1.10 ± 0.93 | 1.25 ± 2.02 | 1.75 ± 1.70 | 0.58 ± 0.21 | 0.001 ± 0.0007 |
| min - max | 0.38-15.24 | 0.20-4.54 | 0-10.70 | 0.18-6.85 | 0.28-1 | 0.0002-0.0032 |
| Micro particles (20-200 μm) | | | | | | |
| average ± SD | 0.58 ± 1.49 | 0.19 ± 0.31 | 0.39 ± 1.20 | 0.29 ± 0.45 | 0.53 ± 0.33 | 0.0011 ± 0.0015 |
| min - max | 0-8.97 | 0-1.75 | 0-7.22 | 0-1.87 | 0-1 | 0-0.0066 |
| Nano particles (5-20 μm) | | | | | | |
| average ± SD | 0.71 ± 0.1 | 0.31 ± 0.33 | 0.40 ± 0.70 | 0.47 ± 1.02 | 0.59 ± 0.22 | 0.0007 ± 0.0008 |
| min - max | 0-4.95 | 0-1.57 | 0-3.38 | 0-5.07 | 0-1 | 0-0.0029 |
| Ultra praticles (2-5 μm) | | | | | | |
| average ± SD | 0.55 ± 0.57 | 0.34 ± 0.28 | 0.22 ± 0.36 | 0.67 ± 0.62 | 0.72 ± 0.27 | 0.0037 ± 0.0104 |
| min - max | 0.01-2.34 | 0.02-1.01 | 0-1.59 | 0.06-2.85 | 0.32-4.62 | 0.0003-0.0521 |
| Nano+ultra particles (2-20 μm) | | | | | | |
| average ± SD | 1.11 ± 1.22 | 0.63 ± 0.46 | 0.49 ± 0.82 | 0.99 ± 1.01 | 0.69 ± 0.21 | 0.0011 ± 0.0007 |
| min - max | 0.28-5.49 | 0.14-2.15 | 0.0001-3.34 | 0-4.91 | 0.30-1 | 0-0.0027 |
| Pico particles (< 2 μm) | | | | | | |
| average ± SD | 0.43 ± 0.29 | 0.32 ± 0.28 | 0.10 ± 0.07 | 0.57 ± 0.28 | 0.63 ± 0.31 | 0.0026 ± 0.0035 |
| min - max | 0.06-1.19 | 0-1.10 | 0-0.21 | 0.04-0.95 | 0-1 | 0.0001-0.0136 |

The right panel of Figure 2 shows the proportion of Chl*a* in individual size fractions. The average contribution of chlorophyll *a* in a given size class to the total concentration of Chl*a* for all data is the highest for pico-particles (35 %) and ultra-particles (35 %), while the average contribution of Chl*a* in nano and micro-particle classes is about 15 % each. The range of variability of the contributions of individual size classes in the total Chl*a* changed as follows: micro-particles from 0 to 53 %, nano particles from 0 to 76 %, ultra-particles from 11 to 86 % and pico-particles from 5 to 66 % (second part of Table 2). In the GG in the winter, Chl*a* in the ultra+pico particles class had about 50 % share, the rest was for medium and large particles, and it can be seen that despite the small contribution of micro-particles in SPM, the proportion of Chl*a* in this size class was about 20 % for stations Ja1, Milpom and GPU, and for Rewa1 about 40 % (probably the particles lifted from the bottom contained a lot of organic detritus). In turn, for example, for station 92a, the contribution of micro-particles was about 30 %, and the proportion of Chl*a* for the same station is close to zero, which means that these particles were inorganic. In autumn, the contribution of Chl*a* for particles < 5 um (except for the ZN2 station located closest to the mouth of the Vistula River) was on average about 80 %. In the case of OCW waters, an average of > 60 % of Chl*a* in the classes of small particles and about 20 % of micro-particles was observed in winter and spring. In autumn, small particles accounted for more than 80 % of Chl*a* in OCW waters. The micro and nano particles in these waters were mostly inorganic. This also results from SPM analyses. At the Plat1 station, no Chl*a* contribution was observed for the micro particle class and < 20 % Chl*a* contribution in the nano-particle fraction. At the Sopot Pier station, the proportion of Chl*a* in the classes of small particles, < 5 μm, in winter was > 60 % with a maximum value of 95 % observed on SF02. In the spring, the contribution of Chl*a* in the class of small particles for SF04 and SF07 was about 30 %, for SF05 about 65 %, and for SF06 100 %. In the summer season, the contribution of small particles < 5 μm was still dominant, except for experiments SF13 and SF14, during which phytoplankton blooms were observed and the contribution of medium and large particles was > 50 %. In early autumn (SF16), despite a significant contribution of medium and large particle classes in the SPM, it can be seen that the share of Chl*a* for these classes was negligible, while ultra and pico particles had the largest contribution.

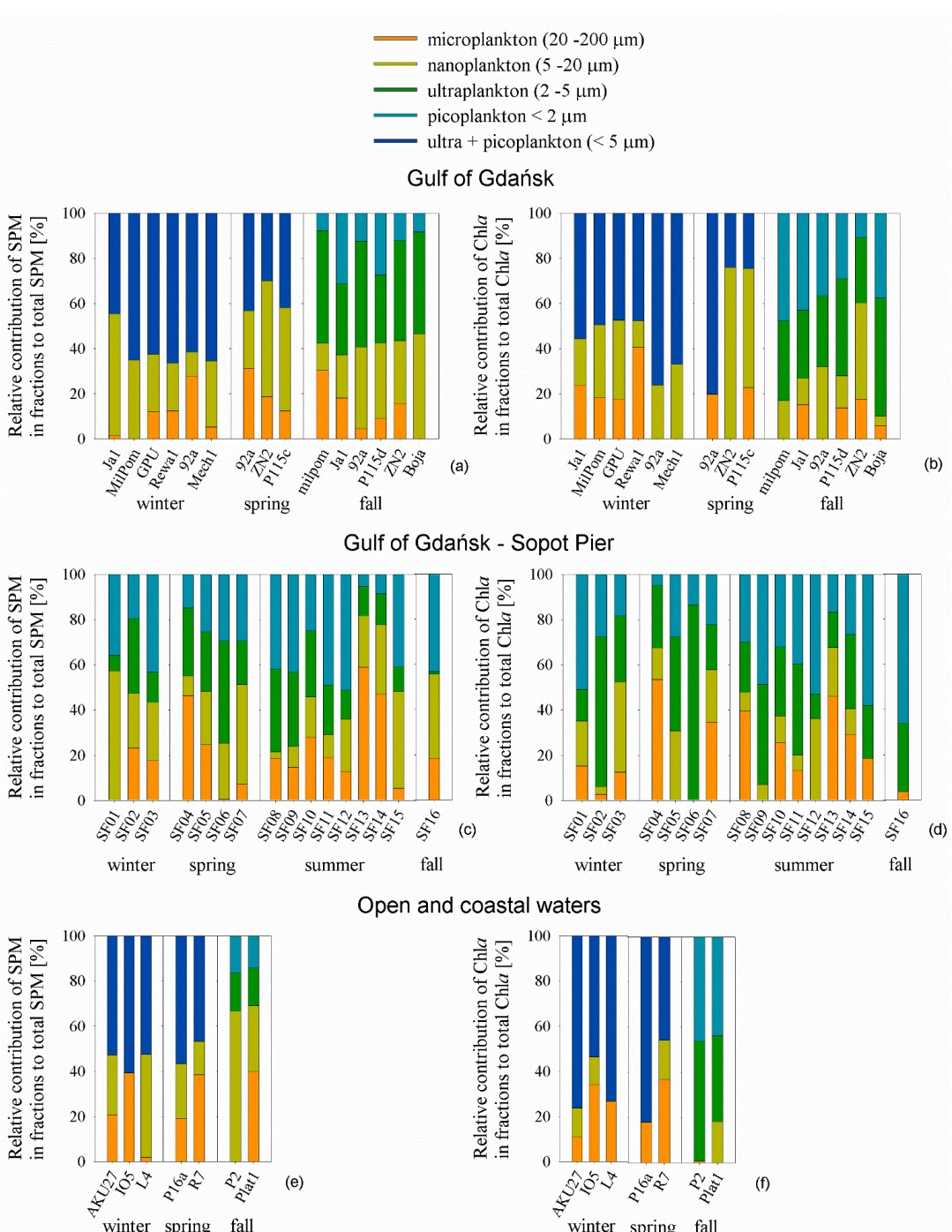

**Figure 2: Relative contribution of SPM in the selected fraction to the total SPM – left panel (a, c, e), and Chl*a* in the selected fractions to the total Chl*a* – right panel (b, d, f), divided due to region (Gulf of Gdańsk, Sopot Pier, Open and coastal waters) and seasons.**

**Table 2: Contributions of particles from different size classes (micro, nano+ultra, nano, ultra, and pico_ to the total SPM and total Chl*a* (n=38). The mean values ± standard deviation (SD) and the variability range are given for all data and in division on sampling area.**

| | all data | Gulf of Gdańsk | Sopot Pier | Open and coastal waters |
|---|---|---|---|---|
| $SPM_{micro}/SPM$ | 17.2 % ± 14.3 % | 13.3 % ± 10.1 % | 21.3 % ± 16.3 % | 22.8 % ± 16 % |
| | 0-58.9 % | 0-31.2 % | 0-58.9 % | 0-39.9 % |
| $SPM_{nano}/SPM$ | 28.3% ± 15.6 % | 31.6 % ± 13 % | 25.4 % ± 14.1 % | 29.5 % ± 19.9 % |
| | 0-66.7 % | 10.8 % - 53.9 % | 3.1 % - 57.3 % | 0- 66.7 % |
| $SPM_{ultra}/SPM$ | 26 % ± 13.9 % | 41.2 % ± 7.7 % | 21.7 % ± 11.8 % | - |
| | 1.1 % -49.9 % | 30 % -49.9 % | 1.1 % - 45.2 % | - |
| $SPM_{pico}/SPM$ | 27 % ± 14.3 % | 16.6 % ± 9.3 % | 31.6 % ± 13.8 % | - |
| | 5.1 % -51.3 % | 7.8 % - 31.3 % | 5.1 % - 51.3 % | - |
| $SPM_{pico+ultra}/SPM$ | 50 % ± 16.8 % | 53.4 % ± 12.8 % | - | 46.9 % ± 4.6 % |
| | 0-66.5 % | 30 % -66.5 % | - | 45.9 % - 60.7 % |
| $Chla_{micro}/Chla$ | 15.8 % ± 14.9 % | 13 % ± 11.5 % | 18.4% ± 17.3 % | 18.2 % ± 14 % |
| | 0-53.4 % | 0-40.5 % | 0-53.4 % | 0-36.9 % |
| $Chla_{nano}/Chla$ | 18% ± 6.6 % | 27.2 % ± 19 % | 14.7 % ± 12.4 % | 8.6 % ± 7.8 % |
| | 0-75.8 % | 0.2 % - 75.8 % | 0- 40% | 0-18.3 % |
| $Chla_{ultra}/Chla$ | 35.3 % ± 16.6 % | 36.8 % ± 8.3 % | 33.4 % ± 19.1 % | - |
| | 10.7 % -86.5 % | 29.1 % - 52.2 % | 10.7 % - 86.5 % | - |
| $Chla_{pico}/Chla$ | 34.7 % ± 15.4 % | 34.2 % ± 12 % | 33.5 % ± 17 % | - |

| | | | |
|---|---|---|---|
| | 5 % - 66.1 % | 10.7 % - 47.7 % | 5 % - 66.1 % | - |
| Chl$a$ $_{pico+ultra}$/ Chl$a$ | 53.6 % ± 22.9 % | 52.5 % ± 18.8 % | - | 73.2 % ± 14 % |
| | 0-82.1 % | 24.2 % - 80.2 % | - | 45.8 % - 82.1 % |

The POM/SPM ratio expresses the proportion of organic matter in the total mass of suspended solids and can vary from 0 (purely mineral suspension) to 1 (100 % organic suspension). The dataset was divided into 5 classes, reflecting the change in suspension composition in the analyzed dataset. POM/SPM for the original samples ranged from 0.28 to 1 (Figure 3).

Analyzes of the POM/SPM ratio in individual size classes show similar variability. It can be seen that with an increase in the SPM value, the proportion of organic particles in the suspension decreases. The exception is the variation of POM/SPM for ultra and pico particles. Here, the contribution of organic matter increases with increasing SPM. The average contribution of POM/SPM in our data set was about 60 % for the entire suspension, as well as for the micro and nano particle classes. On the other hand, for classes of particles with diameters below 5 μm, the average contribution of POM/SPM was higher, > 60 %. In

Figure 3, for all data, it can be seen that the winter season, regardless of the place where POM/SPM samples were taken, assumes the lowest values from SPM . No trends were observed between the remaining seasons and sampling sites. In the case of micro particles, most of the GG samples in autumn are dominated by inorganic particles (POM/SPM < 25 %). For nanoparticles, in the winter, inorganic matter dominated, and in the autumn, organic matter dominated. For ultra particles, no seasonal and spatial dependencies of POM/SPM vs SPM are visible. On the other hand, for pico particles, we observed that

POM/SPM increases with the increase in SPM: in winter POM/SPM had the lowest values, then in spring and autumn it was on average 65 % and the highest values reached in summer on average 80 %.

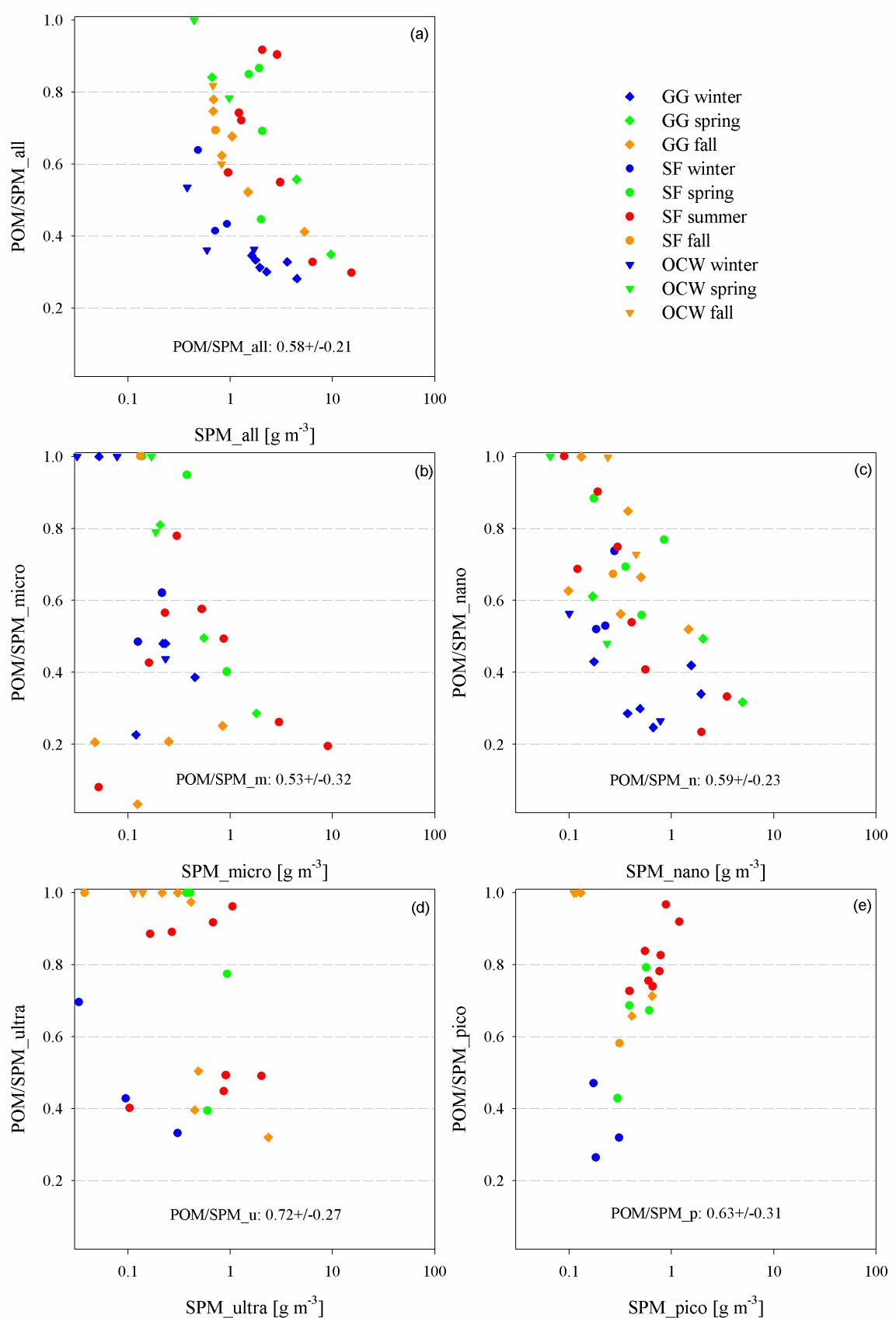

**Figure 3: Relationships between the POM/SPM ratio and the SPM concentration for the original water samples and in the size classes: micro, nano, ultra and pico. Mean values ± standard deviation are shown in the graph. Markers shapes and colours distinguish the season and sampling site (GG - Gulf of Gdańsk, SF - Sopot Pier, OCW - open and coastal waters).**

The Chl$a$/SPM ratio, which is an indicator of the variability of the photosynthetic contribution of live plankton in the entire suspension population, ranges from $6*10^{-4}$ to $4.1*10^{-3}$ for the original samples. For earlier studies of the Baltic Sea, Woźniak et al. (2022) had a variation of $1*10^{-4}$ to $9.3*10^{-3}$. In individual size classes, the variability is similar and with increasing proportion of live phytoplankton in the entire suspension, the proportion of POM/SPM increases. For micro-size Chl$a$/SPM it varies from $6*10^{-5}$ to $9.5*10^{-3}$, for nano-size from $9*10^{-6}$ to $4.6*10^{-3}$, ultra-size from $6*10^{-4}$ to $6*10^{-3}$, and pico-size from $3*10^{-4}$ to $6.4*10^{-3}$.

## 3.2 Variability of absorption properties of suspended and dissolved matter (all particles, detritus, phytoplankton and CDOM)

Figure 4 shows the spectra of light absorption coefficients for all suspended particles, detritus, and phytoplankton determined in the analyzed dataset. The left panel (Figure 4a, c, e) shows the variability of these coefficients for the original unfractionated seawater samples. For an analyzed data sat variability of two orders of magnitude was recorded. The bold line shows the average absorption spectra. The range of variation of absorption coefficients for selected wavelengths is shown in the form of boxes with whiskers. The boxes show the data set between the 25th and 75th percentiles, while the whiskers indicate the 10th and 90th percentiles. Points outside the whiskers are outliers. The line in the boxes indicates the median. The selected wavelengths correspond to the mid-wavelengths of the bands observed by the OLCI (Ocean and Land Colour Instrument) sensor on the Sentinel satellite. The right panel of Figure 4 (b, d, f) shows the variability of $a_p(\lambda)$, $a_d(\lambda)$ and $a_{ph}(\lambda)$, calculated for the micro, nano, ultra and pico-size classes. Bold lines are average spectra in a given size class, thin dashed lines show the range of variability (min-max). The variability was greater than two orders of magnitude, whereas for the pico class it was slightly more than one order of magnitude.

Figure 4g shows the spectra of the coefficients of light absorption by chromophoric dissolved organic matter (CDOM) calculated for the analyzed dataset. The average slope of the spectra ($S_{300-600}$) determined for the 300-600 nm range was 0.022 ± 0.001 m$^{-1}$. .In GG and Sopot Pier $S_{300-600}$ was 0.022+/-0.001, in OCW: 0.024+/-0.001. The light absorption coefficients: $a_{ph}(\lambda)$, $a_d(\lambda)$ and $a_{CDOM}(\lambda)$, allowed one to calculate the light absorption budget for the selected wavelength of 443 nm (Figure 4h). For the analyzed dataset, the average contribution of light absorption by phytoplankton was 29 % ± 14 %, the average contribution of light absorption by detritus was 19 % ± 9 % (the average proportion of $a_{ph}(443)$, $a_d(443)$ and $a_{CDOM}(443)$ is marked with a red triangle). The greatest contribution to the total light absorption was made by CDOM: 52 % ± 20 %. If we take into account the division into sampling areas, in the GG the average contribution of light absorption by phytoplankton was 27 % ± 8 %, the average contribution of detritus: 22 % ± 8 %, and CDOM: 52 % ± 14 %. Similar proportions of light absorption by phytoplankton, detritus and CDOM were in Sopot Pier (respectively: 31 % ± 14 %, 19 % ± 6 % and 50 % ± 15

%). On the other hand, in OCW the proportions of light absorption by individual sea water components were as follows: 19 % ± 4 %, 12 % ± 7 % and 69 % ± 11 %.

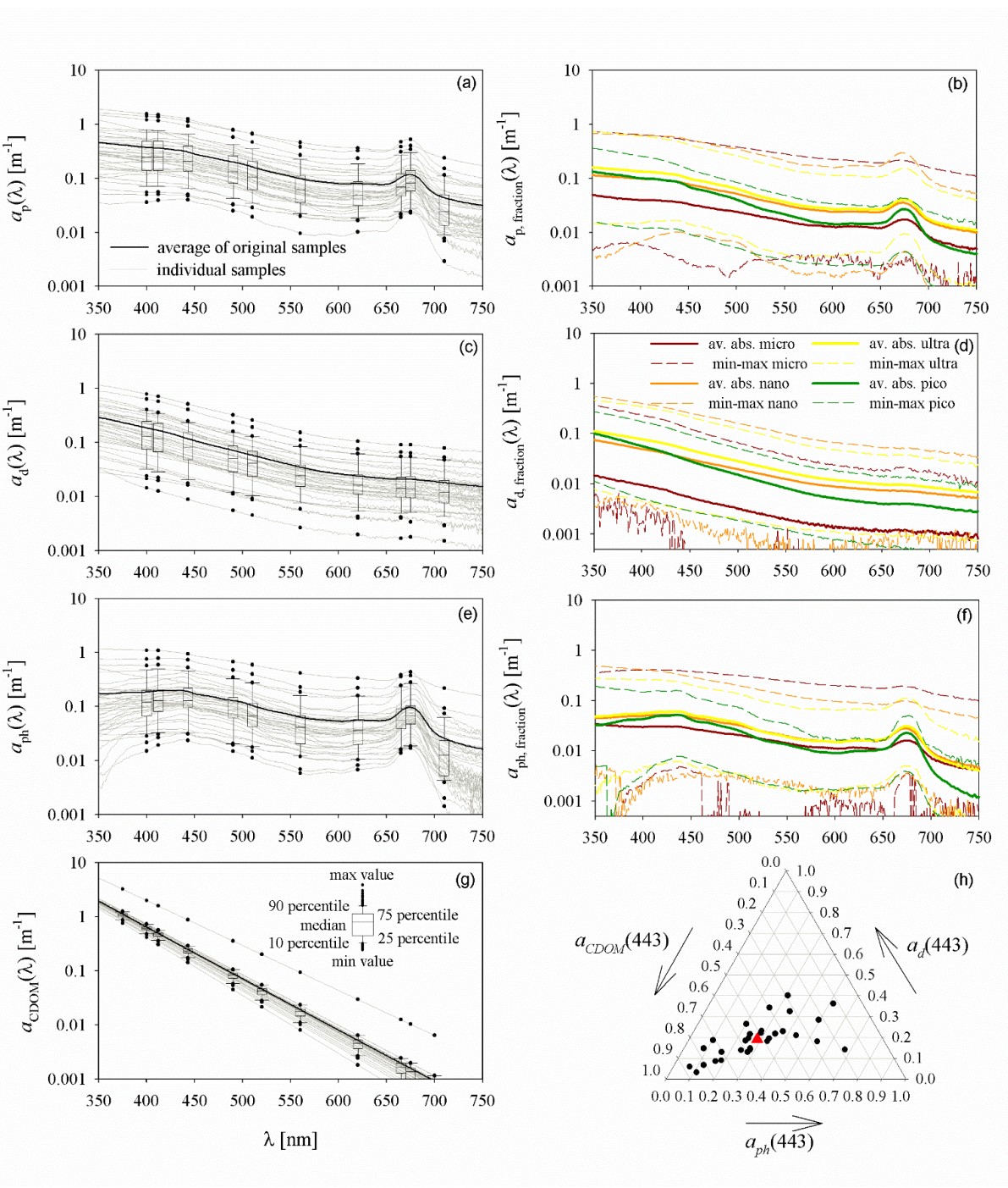

**Figure 4: Spectra of light absorption coefficients by particles suspended in seawater ($a_p(\lambda)$), detritus ($a_d(\lambda)$) and phytoplankton ($a_{ph}(\lambda)$) for original, unfractionated water samples - left panel (a, c, e) and calculated for micro, nano, ultra and pico-size classes - right panel (b, d, f); spectra of CDOM light absorption coefficients ($a_{CDOM}(\lambda)$) (g) and ternary plot showing the light absorption budget for the 443 nm wavelength in the analyzed dataset (h) (the red triangle shows the average contribution of individual absorption coefficients to the total light absorption).**

### 3.3 Variability of light absorption coefficients vs Chl$a$ and SPM

In this work, we focus on showing the variability of light absorption coefficients by size fractions of particles suspended in water. Therefore, it is important to show how these different size fractions depend on the concentration of Chl$a$ and SPM, which are the basic characteristics of these suspensions.

Figure 5 presents the light absorption coefficients of all particles suspended in seawater for a wavelength of 443 nm depending on the concentration of Chl$a$ and SPM, for the original (unfractionated) seawater samples and for the micro, nano, ultra and pico-size classes in log-log scale. The data was divided due to the season and place of sampling. Approximations are shown for all samples in power form ($y=Ax^B$). It can be seen that the better relationships $a_{p,i}(443)$ (where $i$ means: all – all particles, m – particles of the micro-size class: 20-200 μm, n – particles of the nano class: 5-20 μm, u – from the ultra class: 2-5 μm, and p - particles of the pico class: < 2 μm) were obtained from the SPM than Chl$a$ for all coefficients. The coefficient of determination, $R^2$, for the dependence of $a_{p,all}(443)$ vs Chl$a$ was 0.74 and for the dependence of $a_{p,all}(443)$ vs SPM – 0.85. For the micro, nano and ultra-size classes, $R^2$ for both dependencies was 0.46 and 0.81, 0.73 and 0.88, 0.55 and 0.67, respectively. Only the dependencies $a_{p,p}(443)$ vs Chl$a$ and vs SPM had comparable $R^2$: 0.48 and 0.45. The relationships $a_p(443)$ vs Chl$a$ allow to distinguish the winter season from the summer season (blue and red points). In the winter season, the absorption values increase much faster with the increase of Chl$a$ than in other seasons. Taking into account the relationships between $a_p(443)$ and Chl$a$ in individual size classes, the values are slightly dispersed, but there is also a difference between seasons. In the case of dependence on SPM, a division into seasons is also visible. For all particles in the winter and spring season abs coefficients increase slightly faster with increasing SPM than in summer and autumn. Individual size classes also show seasonal trends, however, further research is necessary to draw clear conclusions. As for the sampling area, as expected, OCW are characterized by lower concentrations of Chl$a$ and SPM than GG, and the related lower values of absorption coefficients.

The Baltic Sea is characterized by a large influence of anthropogenic factors on the optical properties of its waters, including the inflow of a large amount of dissolved and suspended organic substances with river waters into its catchment area (especially the Gulf of Gdańsk, which is strongly influenced by the waters of the Vistula). These are waters with complex optical properties that do not depend solely on Chl$a$, especially in the case of detritus. However, in order to compare, we showed how $a_d$ vs Chl$a$ dependencies look like. Figure 6 shows the light absorption coefficients by detritus for a wavelength of 443 nm depending on the Chl$a$ and SPM concentrations for the original seawater samples and for the micro, nano, ultra and pico size classes. As in the case of $a_{p,i}(443)$, it can also be seen that in most cases better relationships were obtained with the SPM than with the Chl$a$. In the case of size classes, the coefficients of determination $R^2$ for the dependence $a_{d,i}(443)$ vs Chl$a$

and SPM were, respectively: micro - 0.58 and 0.79, nano - 0.54 and 0.87, ultra - 0.38 and 0.61, pico - 0.05 and 0.51. The exception is the dependence of $a_{d,all}(443)$ vs Chl$a$ and SPM, where $R^2$ was 0.61 and 0.50, respectively.

Figure 7 shows the relationships for the light absorption coefficients by phytoplankton for the original water samples and for the size classes. It is somewhat surprising that for the analyzed dataset it was also observed that the coefficients $a_{ph,i}(443)$ correlate better with the SPM than with the Chl$a$. An exception is the dependence for picoplankton, where the $R^2$ coefficient for the dependence $a_{ph,p}(443)$ vs Chl$a$ was 0.66 and for SPM only 0.15. In the remaining cases, $R^2$ for dependence $a_{ph,i}(443)$ vs SPM was higher than for $a_{ph,i}(443)$ vs Chl$a$ and ranged from approximately 0.02 to 0.29. As in the case of light

absorption by all particles, seasonal variation can be observed, both for Chl$a$ and SPM dependence. The winter season is characterized by low Chl$a$ values and relatively low $a_{ph}(443)$. The spring season is characterized by the greatest range of Chl$a$ variability. Summer and autumn are characterized by Chl$a$ values >1 and $a_{ph}(443)$ values highest in a year. Similar trends are visible in the case of division into size classes, with a large sample dispersion for micro particles. The dependence of $a_{ph}(443)$ on SPM is characterized by a high coefficient of determination $R^2 = 0.9$. We can distinguish the winter and spring seasons,

which in most cases lie under the approximation curve, while the data collected in the summer and autumn seasons lie above this curve (see Figure 7).

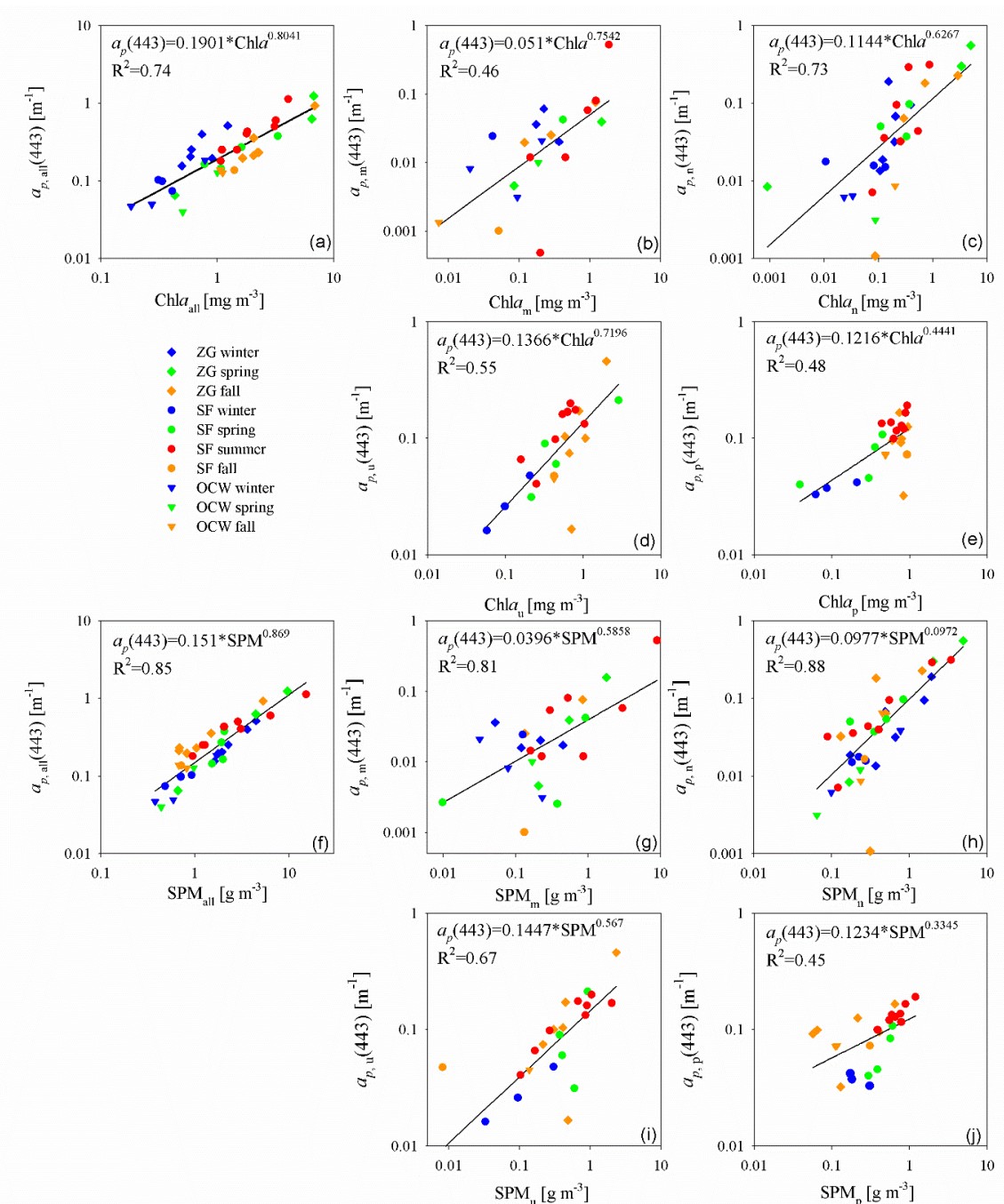

**Figure 5: Relationships of the light absorption coefficients by all unfractionated particles (a, f) and in size classes: micro (b, g), nano (c, h), ultra (d, i) and pico (e, j) from the Chl$a$ (a-e) and SPM (f-j), for the selected wavelength of 443 nm. Note that graphs have different axis scales.**

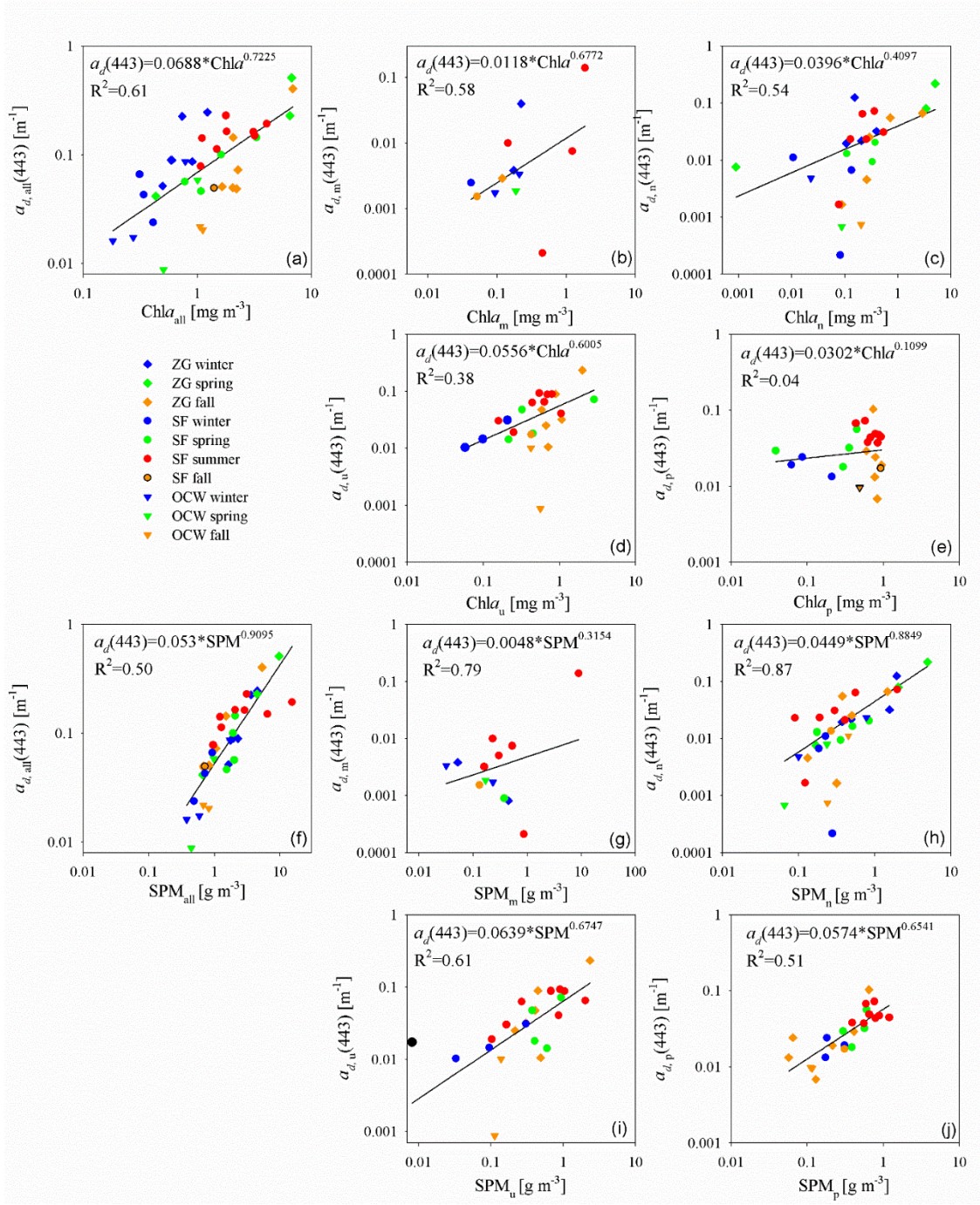

**Figure 6: Relationships of the light absorption coefficients by all unfractionated detritus (a, f) and in size classes: micro (b, g), nano (c, h), ultra (d, i) and pico (e, j) from the Chl$a$ (a-e) and SPM (f-j), for the selected wavelength of 443 nm. Note that graphs have different axis scales.**

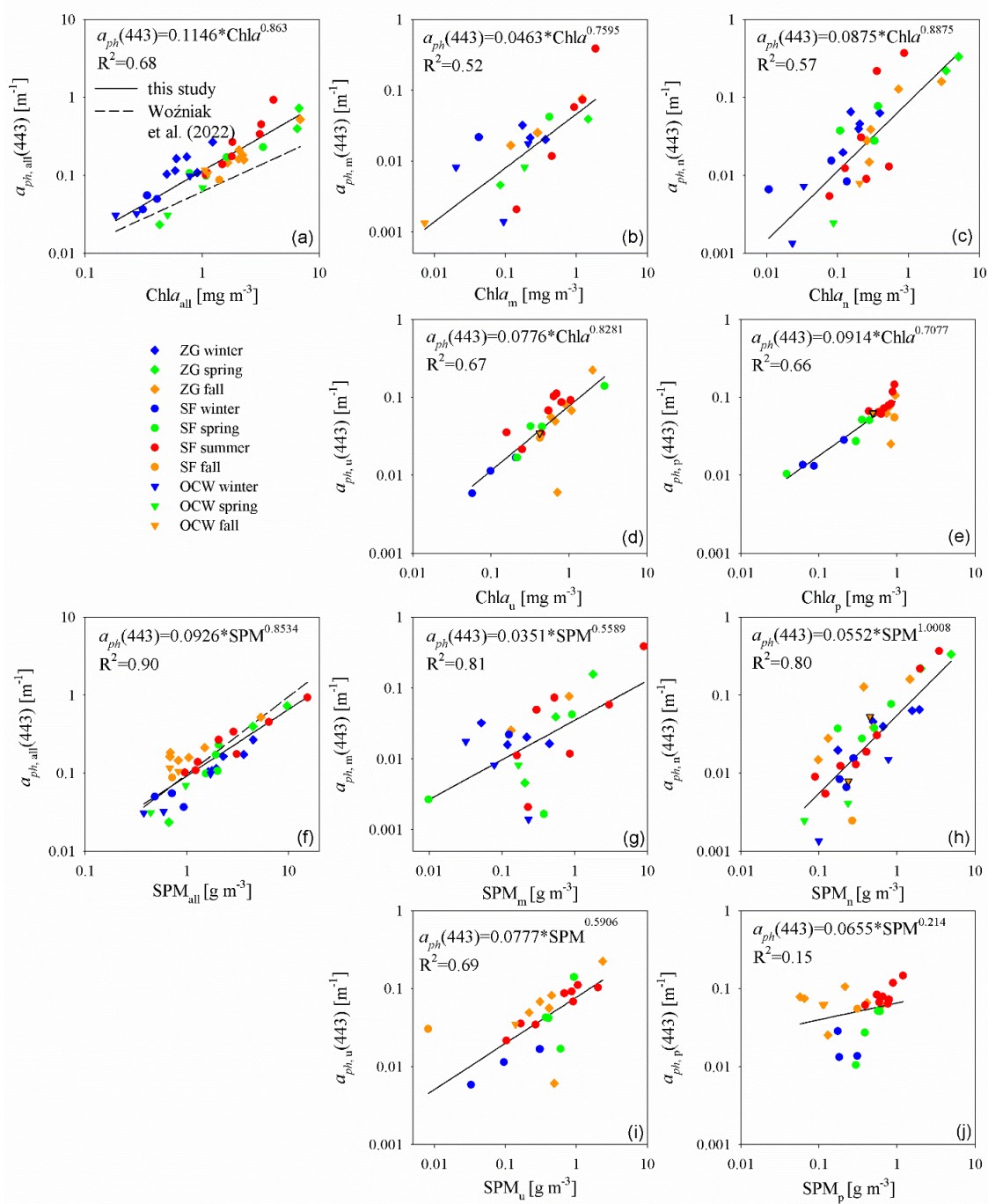

**Figure 7: Relationships of the light absorption coefficients by all unfractionated phytoplankton (a, f) and in size classes: micro (b, g), nano (c, h), ultra (d, i) and pico (e, j) from the Chl*a* (a-e) and SPM (f-j), for the selected wavelength of 443 nm. Note that graphs have different axis scales.**

**3.4 Contribution of size classes to the total light absorption by all particles, detritus and phytoplankton for the wavelength of 443 nm**

Measurements of the light absorption coefficients by particles, including detritus and phytoplankton, provided information on the contribution of the particle size classes (micro, nano, ultra and pico) to the total light absorption coefficients. Table 3 contains such contributions in our dataset (all data, and divided for sampling area: GG, SF, OCW) divided into 4 size classes according to Ciotti et al. (2002):: micro i.e. large particles (20-200 μm), nano-particles (2-20 μm) , ultra-particles (2-5 μm) and pico-particles (<2 μm). The largest contribution has medium particles with sizes from 2 to 20 μm - for each case, on average above 50 %. The small particles in the total absorptions had an average contribution of about 40 %. For all data it can be seen that in the total light absorption by particles, detritus or phytoplankton, particles with sizes < 5 μm - i.e. pico-particles and ultra-particles, had the largest contributions (on average about 38 % and 31 %). Particles with a size of 5-20 μm accounted for approximately 20 % of all particles and phytoplankton, and 29 % of the detritus. The rest was due to the contribution of large particles, about 5-10 % on average. A significant contribution of micro particles to the absorption is observed only for the summer phytoplankton bloom (SF13), and for SF03 and SF04 (end of winter and spring) and autumn bloom at station R7. Graphically, the contribution of size classes to total absorption is illustrated in Figure 8 with division into sampling areas: the Gulf of Gdańsk and a separate set of data from Sopot Pier, as well as open and coastal waters outside the GG. In addition, the data were compiled according to the sampling season. The average contribution in $a_p(443)$ from GG of micro particles was 6.6 %, nano particles: 24.8 %, ultra particles: 42.3 % and pico particles: 26.3 %. The average proportion of $a_p(443)$ from SF was respectively for micro: 9.5 %, for nano: 20.5 %, for ultra: 32.8 % and pico: 37.2 %. For OCW waters, the average contributions in $a_p(443)$ of particles from the micro, nano, ultra and pico size classes were: 8.7 %, 16.9 %, 34.2 % and 40.1 %, respectively. In the case of $a_d(443)$ for micro particles, the smallest contribution was recorded in GG (average 1.7 %), while in SF and OCW it was 6.3 % and 7.4 %, respectively. The proportion of nano particles was higher on average 30 % in all regions. In $a_d(443)$, ultra particles dominated in each area of the study: in GG an average of 53 %, in Sopot Pier 39 % and OCW 41 %, and pico particles: in SF 38 % and OCW 35.7 %, and in GG 26.6 %. There is greater regional variation in uptake by phytoplankton. The smallest contribution of micro particles was observed in GG 9.7 %, and in OCW 11 %, in SF 12 %. Nano particles had the greatest contribution in $a_{ph}(443)$ in GG (27.4 %) and the smallest in OCW (15.6 %). Ultra particles also had the highest contribution in $a_{ph}(443)$ in GG (37.6 %) and about 30 % in Sopot Pier and OCW. In turn, the largest contribution of pico particles was observed in OCW (42.7 %) and the smallest in GG (25.3 %).

To show the contribution of phytoplankton to the light absorption by all particles, the aph(443)/ap(443) ratio was analyzed for the original samples and in size classes. The average $a_{ph}(443)/a_p(443)$ for unfractionated samples was 62 % ± 12 % (36 % - 84 %). In the case of the micro size class, the average $a_{ph}(443)/a_p(443)$ ratio was 83 % ± 26 % (0-100 %), in the nano class - 50 % ± 29 % (0-100 %), and in the ultra and pico classes - 53 % ± 12 % (34 % - 80 %) and 60 % ± 16 % (26 % - 87 %), respectively. Thus, it can be seen that in the analyzed data set, the proportion of phytoplankton absorption in total absorption by all suspended particles was significant. However, in the case of nano and ultra-size classes, there are many cases where detritus has a dominant contribution in light absorption.

**Table 3: Contributions of particles from different size classes (micro, nano+ultra, nano, ultra, and pico) to the total light absorption by particles, detritus and phytoplankton for a wavelength of 443 nm (n=38). The mean values ± standard deviation (SD) and the variability range are given for all data and in division on sampling area.**

| | all data | Gulf of Gdańsk | Sopot Pier | Open and coastal waters |
|---|---|---|---|---|
| $a_{p,micro}/a_p$ | 8.2 % ± 9.6 % | 6.6 % ± 4.4 % | 9.5 % ± 12.7 % | 8.8 % ± 9.1 % |
| | 0-46.9% | 0-12.7% | 0-45.9 % | 0-25.3 % |
| $a_{p,nano}/a_p$ | 21.6 % ± 16.4 % | 24.8 % ± 21 % | 20.6 % ± 11.2 % | 16.8 % ± 13.1 % |
| | 0-78.9 % | 0-78.9 % | 2.8 % - 48.2 % | 6.8 % - 47 % |
| $a_{p,ultra}/a_p$ | 36.8 % ± 13.5 % | 42.3 % ± 13.5 % | 32.7 % ± 11 % | 34.3 % ± 14.6 % |
| | 0-58.6 % | 7.2 % -58.6 % | 15 % - 56.3 % | 0-49.1 % |
| $a_{p,pico}/a_p$ | 33.4 % ± 13.9 % | 26.3 % ± 12.8 % | 37.2 % ± 13.1 % | 40.1 % ± 11.1 % |
| | 10 % -57.6 % | 10 % - 54 % | 10.3 % - 56.8 % | 27 % - 57.6 % |
| | | | | |
| $a_{d,micro}/a_d$ | 4.2 % ± 12 % | 1.7 % ± 4.1 % | 6.3 % ± 17.1 % | 5 % ± 7.4 % |
| | 0-71.9 % | 0-16.1 % | 0-71.9 % | 0-21.1 % |
| $a_{d,nano}/a_d$ | 29.5 % ± 22.7 % | 29.6% ± 24.3 % | 28.7 % ± 20.5 % | 30.8 % ± 23.9 % |
| | 0-87.9 % | 0.1 % – 87.9 % | 1.2 % - 80.8 % | 0.1 % - 82.4 % |
| $a_{d,ultra}/a_d$ | 44.9 % ± 16.2 % | 52.9 % ± 18.1 % | 39 % ± 9.7 % | 41.1 % ± 16.6 % |
| | 4 % -93.4 % | 14.5 % - 93.4 % | 24.4 % - 54 % | 4 % - 58.7 % |
| $a_{d,pico}/a_d$ | 33.1 % ± 10.9 % | 26.6 % ± 9.7 % | 38 % ± 9.6 % | 35.7 % ± 8.5 % |
| | 9.4 % -55.9 % | 9.4 % - 47.5 % | 22.7 % - 55.9 % | 25.1 % - 47 % |
| | | | | |
| $a_{ph,micro}/a_{ph}$ | 8.8 % ± 11.5 % | 9.7 % ± 7.4 % | 12.1 % ± 15 % | 10.9 % ± 11.4 % |
| | 0-41.7 % | 0-21.5 % | 0-41.7 % | 0-26.6 % |
| $a_{ph,nano}/a_{ph}$ | 18.2% ± 18.3 % | 27.4 % ± 20.9 % | 21 % ± 14.6 % | 15.6 % ± 13.6 % |
| | 0-80.3 % | 0-80.3 % | 2.7 % - 48.2 % | 4.4 % - 45.5 % |
| $a_{ph,ultra}/a_{ph}$ | 26.6 % ± 18 % | 37.6 % ± 14 % | 29.4 % ± 12.8 % | 30.8 % ± 13.2 % |
| | 0-63.7 % | 3.8 % - 63.7 % | 11.1 % - 60.4 % | 0-41.1 % |
| $a_{ph,pico}/a_{ph}$ | 27.2 % ± 20.5 % | 25.3 % ± 15.1 % | 37.5 % ± 17.9 % | 42.7 % ± 12.4 % |
| | 0-62.9 % | 6.8 % - 57.9 % | 7.7 % - 62.9 % | 24.6 % - 59.6% |

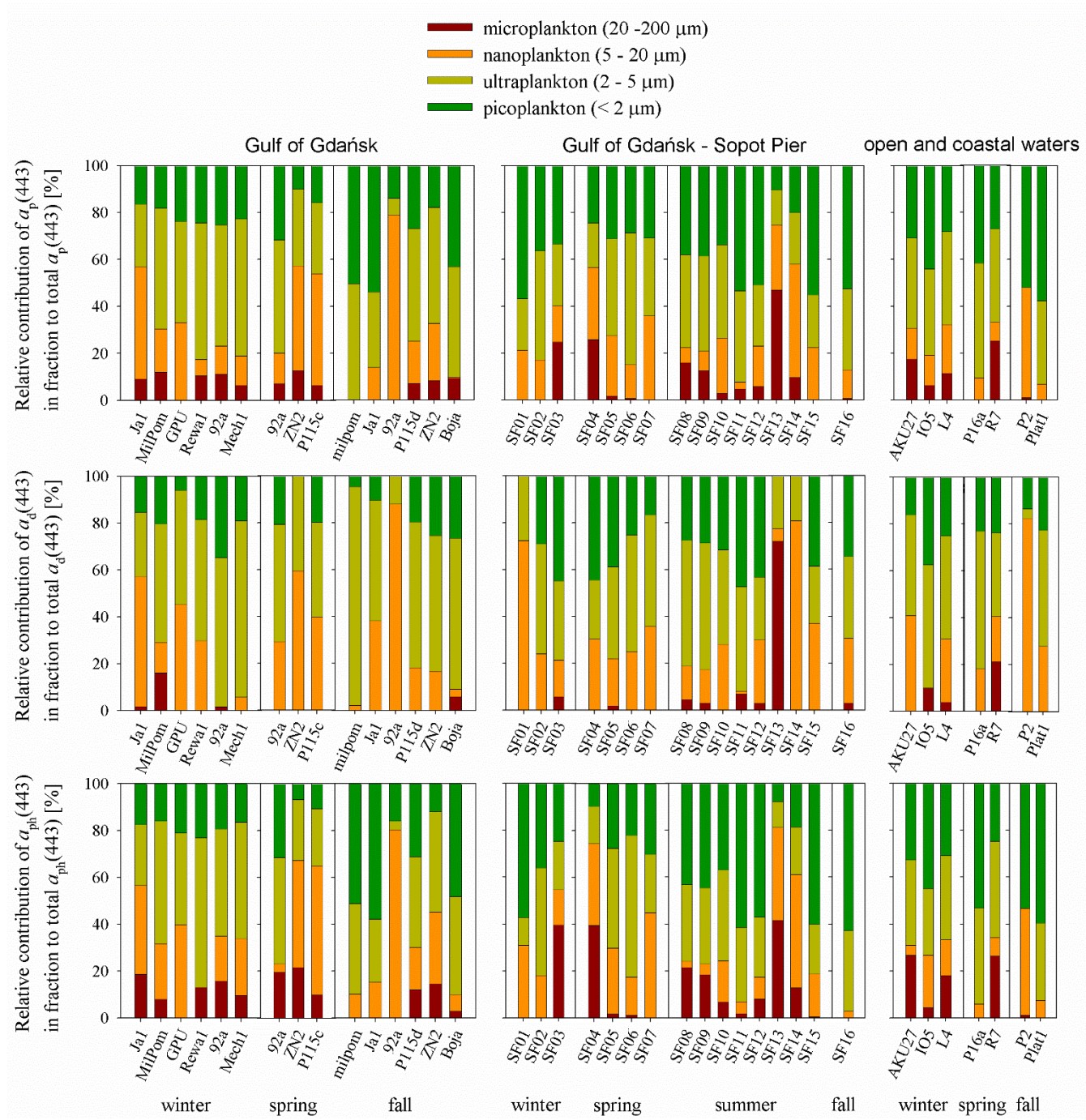

**Figure 8: Contribution of particles from size classes in the total light absorption by all particles, detritus and phytoplankton according to the division into 4 size classes according to Ciotti et al. (2002) (micro-, nano-, ultra- and pico-particles) in GG, SF and OCW in for seasons**

465

## 3.5 Average specific light absorption coefficients of particles for distinguished size classes

The conducted analyzes allowed one to determine the average values of chlorophyll-specific light absorption coefficients by particles, detritus, and phytoplankton in size classes. Such average values are often sought because they allow for a simple description of the relationship between biogeochemical and optical quantities. The averages were determined for the data that met the condition of the so-called fraction dominance (Ciotti et al., 2002; Kheireddine et al., 2018), where the ratio of Chl*a*_fraction to Chl*a* concentrations for the original sample was greater than 40% for one size classes. In the event that 2 fractions met this condition simultaneously, the following condition had to be met: the Chl*a*_fraction/Chl*a* ratio had to be greater than 50%. To determine the average values of the mass-specific light absorption coefficients $a_p$, $a_d$, and $a_{ph}$, a similar SPM_fraction dominance criterion to the SPM of the original sample was used. The results are shown in Figure 9 for the classic division into three size classes: micro-size (20-200 μm) - large particles, nano+ultra-size (2-20 μm) - medium particles and pico-size (<2 μm) - small particles. The left panel shows the chlorophyll-specific coefficients $a_p^{(\text{Chl}a)}(\lambda)$, $a_d^{(\text{Chl}a)}(\lambda)$ and $a_{ph}^{(\text{Chl}a)}(\lambda)$, while the right panel shows the mass-specific coefficients $a_p^{(\text{SPM})}(\lambda)$, $a_d^{(\text{SPM})}(\lambda)$ and $a_{ph}^{(\text{SPM})}(\lambda)$. The average $a_p^{(\text{Chl}a)}(\lambda)$ coefficients determined for the three particle size classes have similar values, slight differences can be observed in the spectral range of 500 - 680 nm, where $a_p^{(\text{Chl}a)}(\lambda)$ for the medium particle class has lower values than for the large classes and small particles. With the standard deviations determined for each mean taken into account, it is impossible to unambiguously separate the individual absorption spectra to determine which particle size class it may belong to. In the case of the average $a_p^{(\text{SPM})}(\lambda)$ coefficients, there is a clear difference between the average determined for the micro particle class, which, together with the standard deviations, varies from the average ± SD for the medium and small particle classes. The average $a_d^{(\text{Chl}a)}(\lambda)$ and $a_d^{(\text{SPM})}(\lambda)$ coefficients show slightly greater variation. The $a_d^{(\text{Chl}a)}(\lambda)$ means clearly distinguish size classes, but if we take into account the standard deviations, the possibility of qualifying the spectrum to a given size class of particles decreases. As in the case of $a_p^{(\text{SPM})}(\lambda)$, the coefficients $a_d^{(\text{SPM})}(\lambda)$ clearly distinguish the class of large particles from other particles. The average $a_{ph}^{(\text{Chl}a)}(\lambda)$ coefficients together with the standard deviations also do not allow separating the particle size classes clearly from each other, however, there is a clear difference between the mean determined for large particles and the average for medium and small particles. On the other hand, the average coefficients $a_{ph}^{(\text{SPM})}(\lambda)\pm\text{SD}$ make it possible to clearly distinguish the class of micro particles from the rest of the particles with sizes up to 20 μm.

The same analyzes were carried out for four distinguished particle size classes: micro (20-200 μm), nano (5-20 μm), ultra (2-5 μm) and pico (< 2 μm) and the results are summarized in Figure 10. For each particle size class analysis, it was possible to determine the average chlorophyll-specific coefficients $a_p^{(\text{Chl}a)}(\lambda)$, $a_d^{(\text{Chl}a)}(\lambda)$ and $a_{ph}^{(\text{Chl}a)}(\lambda)$, and the average mass-specific coefficients $a_p^{(\text{SPM})}(\lambda)$, $a_d^{(\text{SPM})}(\lambda)$ and $a_{ph}^{(\text{SPM})}(\lambda)$ with standard deviations (left and right panels, respectively, Figure 10). For the analyzed dataset, it is not possible to unambiguously separate the absorption coefficients into size classes. Only in the case of coefficients $a_d^{(\text{Chl}a)}(\lambda)$ it is possible to distinguish the class of micro-particles from ultra-particles in the spectral range above approx. 550nm.

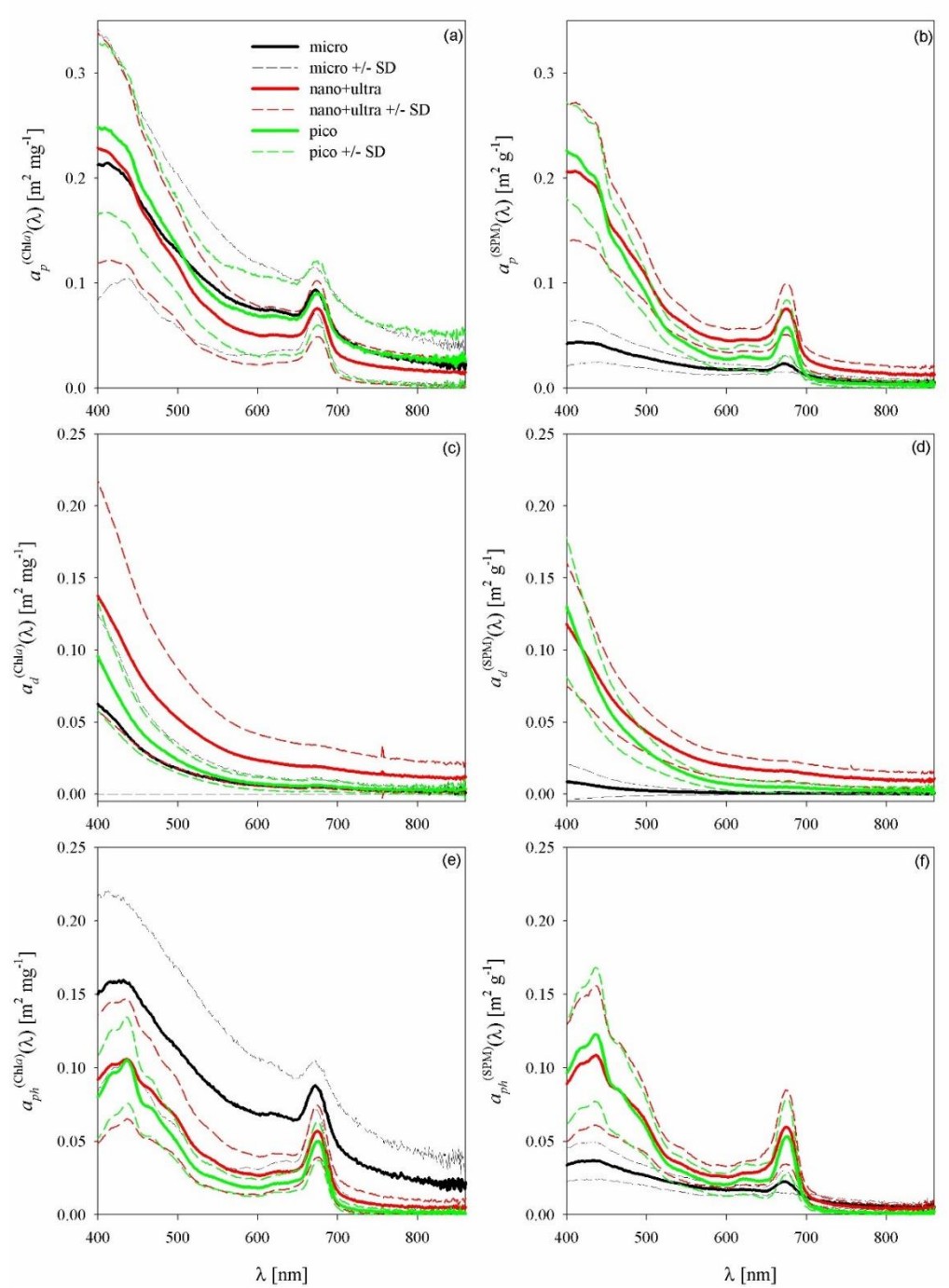

**Figure 9: Mean chlorophyll-specific (left panel) and mass-specific (right panel) light absorption coefficients for all particles (a, b), detritus (c, d) and phytoplankton (e, f) for 3 size classes: micro-, nano-+ultra- and picoplankton. SD means standard deviation.**

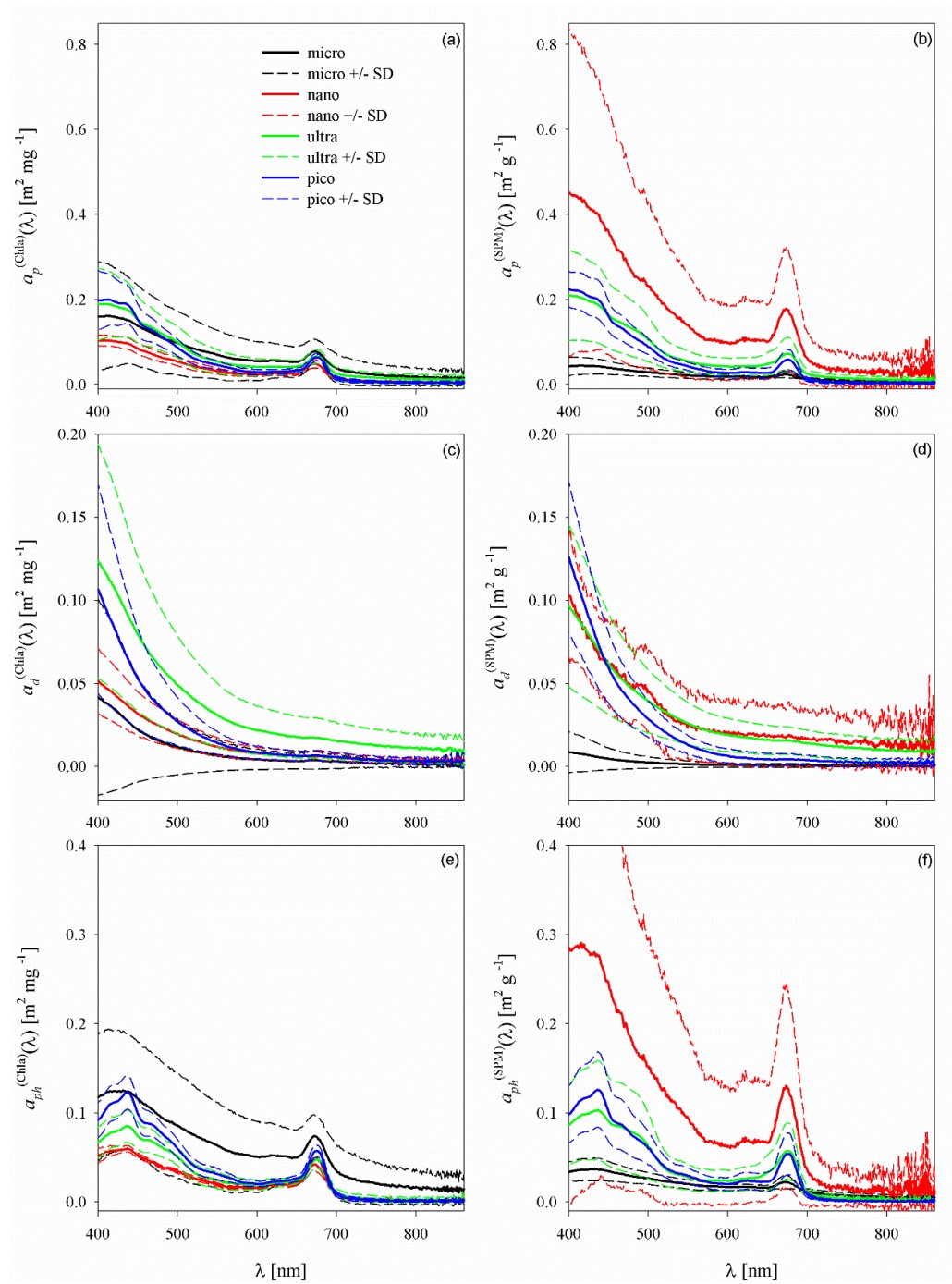

**Figure 10: Mean chlorophyll-specific (left panel) and mass-specific (right panel) light absorption coefficients for all particles (a, b), detritus (c, d) and phytoplankton (e, f) for 4 size classes: micro-, nano-, ultra- and picoplankton. SD means standard deviation.**

505

Table 4 summarizes the average values of specific (chlorophyll and mass) light absorption coefficients for all particles, detritus and phytoplankton calculated for a given size class along with standard deviations for selected wavelengths corresponding to selected bands of the OLCI sensor.

**Table 4: Average values of chlorophyll-specific and mass-specific light absorption coefficients for all particles, detritus and phytoplankton calculated for a given size class with standard deviations for selected wavelengths corresponding to selected bands of the OLCI sensor.**

| | 400 nm | 443 nm | 510 nm | 620 nm | 675 nm | 709 nm |
|---|---|---|---|---|---|---|
| $a_{p,micro}^{(Chla)}(\lambda)$ | 0.160+/-0.135 | 0.147+/-0.098 | 0.091+/-0.072 | 0.056+/-0.041 | 0.077+/-0.028 | 0.033+/-0.032 |
| $a_{p,nano}^{(Chla)}(\lambda)$ | 0.104+/-0.013 | 0.092+/-0.013 | 0.048+/-0.010 | 0.027+/-0.006 | 0.048+/-0.009 | 0.011+/-0.004 |
| $a_{p,ultra}^{(Chla)}(\lambda)$ | 0.190+/-0.085 | 0.167+/-0.061 | 0.086+/-0.035 | 0.041+/-0.018 | 0.066+/-0.016 | 0.021+/-0.015 |
| $a_{p,nano+ultra}^{(Chla)}(\lambda)$ | 0.229+/-0.110 | 0.195+/-0.083 | 0.103+/-0.050 | 0.051+/-0.027 | 0.076+/-0.027 | 0.029+/-0.020 |
| $a_{p,pico}^{(Chla)}(\lambda)$ | 0.244+/-0.055 | 0.199+/-0.038 | 0.082+/-0.018 | 0.031+/-0.004 | 0.061+/-0.007 | 0.011+/-0.004 |
| | | | | | | |
| $a_{d,micro}^{(Chla)}(\lambda)$ | 0.042+/-0.058 | 0.025+/-0.035 | 0.010+/-0.015 | 0.004+/-0.006 | 0.004+/-0.006 | 0.003+/-0.004 |
| $a_{d,nano}^{(Chla)}(\lambda)$ | 0.052+/-0.032 | 0.035+/-0.014 | 0.018+/-0.008 | 0.007+/-0.003 | 0.006+/-0.003 | 0.005+/-0.002 |
| $a_{d,ultra}^{(Chla)}(\lambda)$ | 0.124+/-0.070 | 0.086+/-0.050 | 0.045+/-0.027 | 0.020+/-0.014 | 0.018+/-0.012 | 0.015+/-0.011 |
| $a_{d,nano+ultra}^{(Chla)}(\lambda)$ | 0.138+/-0.080 | 0.093+/-0.056 | 0.048+/-0.032 | 0.021+/-0.017 | 0.019+/-0.015 | 0.017+/-0.014 |
| $a_{d,pico}^{(Chla)}(\lambda)$ | 0.155+/-0.044 | 0.085+/-0.027 | 0.032+/-0.010 | 0.008+/-0.004 | 0.006+/-0.003 | 0.005+/-0.003 |
| | | | | | | |
| $a_{ph,micro}^{(Chla)}(\lambda)$ | 0.118+/-0.071 | 0.122+/-0.064 | 0.081+/-0.058 | 0.052+/-0.036 | 0.073+/-0.023 | 0.030+/-0.028 |

| | | | | | | |
|---|---|---|---|---|---|---|
| $a_{ph,nano}^{(Chla)}(\lambda)$ | 0.052+/-0.008 | 0.057+/-0.003 | 0.031+/-0.003 | 0.019+/-0.002 | 0.041+/-0.0074 | 0.006+/-0.002 |
| $a_{ph,ultra}^{(Chla)}(\lambda)$ | 0.067+/-0.017 | 0.081+/-0.018 | 0.042+/-0.001 | 0.021+/-0.005 | 0.048+/-0.010 | 0.007+/-0.002 |
| $a_{ph,nano+ultra}^{(Chla)}(\lambda)$ | 0.092+/-0.044 | 0.102+/-0.039 | 0.055+/-0.024 | 0.029+/-0.013 | 0.057+/-0.018 | 0.012+/-0.009 |
| $a_{ph,pico}^{(Chla)}(\lambda)$ | 0.089+/-0.019 | 0.114+/-0.019 | 0.049+/-0.011 | 0.023+/-0.002 | 0.055+/-0.006 | 0.006+/-0.002 |
| | | | | | | |
| $a_{p,micro}^{(SPM)}(\lambda)$ | 0.054+/-0.017 | 0.052+/-0.007 | 0.032+/-0.008 | 0.020+/-0.004 | 0.028+/-0.004 | 0.011+/-0.005 |
| $a_{p,nano}^{(SPM)}(\lambda)$ | 0.453+/-0.385 | 0.380+/-0.300 | 0.212+/-0.180 | 0.107+/-0.094 | 0.176+/-0.140 | 0.050+/-0.050 |
| $a_{p,ultra}^{(SPM)}(\lambda)$ | 0.210+/-0.107 | 0.178+/-0.086 | 0.089+/-0.042 | 0.043+/-0.021 | 0.072+/-0.038 | 0.021+/-0.009 |
| $a_{p,nano+ultra}^{(SPM)}(\lambda)$ | 0.0189+/-0.160 | 0.152+/-0.113 | 0.086+/-0.070 | 0.045+/-0.038 | 0.057+/-0.035 | 0.028+/-0.028 |
| $a_{p,pico}^{(SPM)}(\lambda)$ | 0.227+/-0.046 | 0.185+/-0.047 | 0.074+/-0.017 | 0.030+/-0.011 | 0.058+/-0.026 | 0.009+/-0.004 |
| | | | | | | |
| $a_{d,micro}^{(SPM)}(\lambda)$ | 0.026 | 0.016 | 0.007 | 0.003 | 0.003 | 0.002 |
| $a_{d,nano}^{(SPM)}(\lambda)$ | 0.144+/-0.022 | 0.090+/-0.016 | 0.061+/-0.031 | 0.028+/-0.028 | 0.029+/-0.029 | 0.026+/-0.026 |
| $a_{d,ultra}^{(SPM)}(\lambda)$ | 0.090+/-0.049 | 0.067+/-0.033 | 0.036+/-0.017 | 0.018+/-0.009 | 0.016+/-0.008 | 0.014+/-0.008 |
| $a_{d,nan+ultra}^{(SPM)}(\lambda)$ | 0.108+/-0.099 | 0.072+/-0.066 | 0.039+/-0.038 | 0.019+/-0.020 | 0.017+/-0.018 | 0.015+/-0.016 |
| $a_{d,pico}^{(SPM)}(\lambda)$ | 0.130+/-0.048 | 0.069+/-0.026 | 0.026+/-0.009 | 0.006+/-0.004 | 0.005+/-0.004 | 0.004+/-0.003 |
| | | | | | | |
| $a_{ph,micro}^{(SPM)}(\lambda)$ | 0.041+/-0.004 | 0.045+/-0.001 | 0.028+/-0.005 | 0.019+/-0.003 | 0.027+/-0.005 | 0.063+/-0.005 |

| | | | | | | |
|---|---|---|---|---|---|---|
| $a_{ph,nano}^{(SPM)}(\lambda)$ | 0.381+/-0.313 | 0.335+/-0.257 | 0.182+/-0.152 | 0.093+/-0.078 | 0.161+/-0.120 | 0.037+/-0.029 |
| $a_{ph,ultra}^{(SPM)}(\lambda)$ | 0.086+/-0.045 | 0.099+/-0.054 | 0.050+/-0.027 | 0.026+/-0.013 | 0.057+/-0.032 | 0.008+/-0.005 |
| $a_{ph,nano+ultra}^{(SPM)}(\lambda)$ | 0.088+/-0.071 | 0.085+/-0.055 | 0.050+/-0.036 | 0.027+/-0.021 | 0.042+/-0.020 | 0.014+/-0.014 |
| $a_{ph,pico}^{(SPM)}(\lambda)$ | 0.096+/-0.035 | 0.116+/-0.043 | 0.047+/-0.016 | 0.024+/-0.010 | 0.053+/-0.026 | 0.005+/-0.002 |

## 4 Discussion

We have carried out a number of analyzes of the variability of the mass of suspended particles, Chl$a$ concentrations and the absorption properties of suspended particles for various size classes in the waters of the southern part of the Baltic Sea. So far, such analyzes have not been carried out empirically. No similar studies have been reported in the available literature so far. Fractionation based on the size of the particles suspended in the water column was carried out for completely different particle size ranges (<50 μm and >50 μm) and was most often associated with geotrace studies, in which the chemical composition of the suspension was determined (e.g., Lam et al., 2015, 2018; Xiang and Lam, 2020; Yigiterhan et al., 2020). However, in the work of Mohammadpour et al. (2017) optical properties of SPM size fractions in littoral waters of Quebec were studied, but for size classes 0.2-0.4 μm, 0.4-0.7 μm, 0.7-10 μm and > 10 μm. The contribution of particles with sizes > 10 μm did not exceed 17 %.

In our analyses, we showed the seasonal variability of Chl$a$ contributions in size classes for three separate regions of the Baltic Sea: Gulf of Gdańsk, Sopot Pier, and open and coastal waters. This division results from the relationship between the optical properties of waters and their location in relation to river mouths, distance from the shore, or bottom bathymetry. This relationship is conditioned by the changing content of optically active water components and water circulation (Sagan, 2008). Water in the Gulf of Gdańsk was strongly influenced by river waters, as evidenced by the average salinity of 6.4 (0.4 - 7.7). At Sopot Pier, the average salinity was 7.6 (7 - 8.3), and at OCW 7.6 (7.2 -8.0). Water in the Gulf of Gdańsk was strongly influenced by river waters, as evidenced by the average salinity of 6.4 (0.4 - 7.7). At Sopot Pier, the average salinity was 7.6 (7 - 8.3), and at OCW 7.6 (7.2 -8.0). The proportions of Chla obtained by us in the micro, nano, ultra and pico size classes in these regions (without division into seasons) were: GG – 13 %, 27 %, 37 % and 34 %; in SF – 18 %, 15 %, 33 % and 34 %, and in OCW – 18 %, 9 %, 73 % (for ultra+pico, where the fraction could not be separated). The proportions of Chl$a$ in various size classes of phytoplankton were reported in the literature. However, in most cases, this type of research concerned primary production. For the Baltic Sea, such studies have not been carried out so far. Reported by Maranon et al. (2001) contributions of Chl$a$ in fractions up to the total concentration of Chl$a$ (integrated for the water column 0-200 m) in the waters of the Atlantic Ocean showed that in most data the contribution of picoplankton was 60-80 % (average 61 %), nanoplankton (including ultraplankton) was approximately 20 % to 30 % (average 29 %) (from low-production regions to temperate regions), while the

contribution of microplankton varied from < 10 % to 20 % (the share was low in most cases and increased to 20 % in upwelling areas and moderate, especially in spring) (average 9 %). Deng et al. (2022) presented results for data from the southern part of the China Sea from the surface layer, where the average contribution of picoplankton was 81 %, nanoplankton (including ultraplankton) 12 % and microplankton 7 %. In turn, Decembrini et al. (2014) for the Magellan Strait investigated the seasonal variability of the contributions of fractions in total Chl*a* integrated for a depth of 0-50 m. For the spring season, the contribution of picoplankton in total Chl*a* was on average 12 %, and the micro+nano fraction was 88 %. In the summer season, the contribution of picoplankton was on average 60 %, micro+nano 40 %. In late summer and early autumn, the average contribution of picoplankton was 47 % and micro+nano 53 %. Saggiomo et al. (1994) for austral summer (February–March 1991), in the Strait of Magellan, reported that the most important characteristic identified for this area was the confinement of the microplankton fraction to the external parts of the Strait and the rather uniform dimensional structure of the phytoplankton communities (< 5 μm) within the internal sectors. In particular, the nanoplankton fraction (2-10 μm) comprised 33 %, while the picoplankton one (0.5-2 μm) represented 62 % of the total. Cermeno et al. (2005) studied the photosynthetic efficiency of fractionated phytoplankton in the Ria de Vigo (Iberian Penisula). The contribution of fractions with sizes < 5 μm was on average 46 %, fractions with sizes 5-20 μm and > 20 μm were 27 % each, in the total Chl*a* integrated in the euphotic zone (0-20 m). All the cases of phytoplankton fractionated by size described above concern studies related to primary production. The data presented in this paper do not differ from those reported in the literature. By comparison of the above data, it can be seen that the percentage contribution of individual size fractions to the total concentration of Chl*a* is not a constant value, but changes over time and space.

The overall proportion of organic matter in the total POM/SPM suspension for the analyzed dataset averaged 58% for total particles (see Figure 3). In the works of Woźniak and Meler (2020) and Woźniak et al. (2022) on the study of the optical properties of the waters of the Baltic Sea (conducted in the period 2017-2020), the presented POM/SPM ratios were characterized by similar values. The average POM/SPM was 61 % and 62 %, respectively. In previous studies, the average POM/SPM for the Baltic Sea was 80 % (Woźniak et al., 2011 - for the period 2006-2009) and 76% (Meler et al., 2016 a, b - for the period 2006-2012). Long-term variability from previous years indicated a higher proportion of organic particles in the total suspension, while data from 2017-2021 may indicate that the proportion of detrital and mineral particles in the particle composition increased. However, this may be due to the increased number of stations located in the coastal waters zone in the total database. This, in turn, is caused by the variability of the weather (change in climatic conditions). In recent years, optical cruises, which have been taking place more or less at similar times of the year for 20 years, are characterized by more windy weather, which results in sailing closer to the coastal zone (there is less data from open waters), and therefore there is a greater proportion of particles flowing from land and/or lifted from the bottom, as a result of stormy weather. It is also possible that during the period of our research, the influence of river waters from the Vistula was greater than in the case of previous research. Research by Grelowski and Wojewódzki (1996) shows that the flow of the Vistula waters feeding into the GG carry sedimentary material extends horizontally from 2 to 15 nautical miles from river mouth, and has a vertical range of 0.5-12 m,

depending mainly on the wind speed and direction, as well as a combination of factors such as the river water discharge rate, sea level and the duration of their interactions (Matciak and Nowacki, 1995).

The composition of suspension particles and organic solutes has a significant impact on their absorption properties. The absorption budget presented by us for the selected wavelength of 443 nm indicates that phytoplankton absorbed on average 29 % of light, 19 % was due to detritus, and 52 % was due to CDOM (Figure 4h). Similar analyzes were carried out by Woźniak

et al. (2011) and reported for the wavelength of 440 nm the average contribution were: $(a_{ph}(440) + a_d(440)) - 45$ % and $a_{CDOM}(440) - 55$ %. For the same wavelength (443 nm), Figures 5-7 show how the absorption properties of all particles (including detritus and phytoplankton) changed with the change of Chl$a$ and SPM, i.e. the basic characteristics of the suspension. The similar relationships (for unfiltered samples) were slightly different in Woźniak et al. (2011). There, for the wavelength of 440 nm, the coefficient of determination for the dependence $a_{p,all}(440)$ vs Chl$a$ was 0.73, and for the dependence

$a_p(440)$ vs SPM - 0.53. In the case of the coefficient $a_{d,all}(443)$ for the Baltic Sea, it has been already shown that the dependence on SPM is better than the dependence on Chl$a$ (Woźniak et al., 2011; Meler et al., 2017). Comparing these relationships with those obtained by Woźniak et al. (2022), shown for a wavelength of 440 nm, it can be seen that also in this case the $a_{ph}$ vs SPM relationships have higher determination coefficients $R^2$ than the $a_{ph}$ vs Chl$a$ relationships, 0.86 and 0.82, respectively.

The obtained average spectra for the dominant size class do not clearly explain the shape of the absorption spectrum,

regardless of the division into 3 or 4 size classes (Figure 9-10). For the analyzed data set, on average, 52% of suspended matter was organic of both autogenous and allogeneic origin, and the contribution of inorganic matter to light absorption was not significant (Woźniak and Dera, 2007). Coefficent $a_p = a_{ph} + a_d$, therefore $a_p$, $a_{ph}$ and $a_d$ can be all expressed in relation to Chl$a$ and SPM. For clean ocean waters (the so-called Case1) Bricaud et al. (1998) presented Chl$a$-specific non-algal particles absorption $a_d^{(Chla)}$ determined from the difference of $a_p^{(Chla)}$ and $a_{ph}^{(Chla)}$ (Woźniak & Dera, 2007). In the case of the optically

complex Baltic Sea waters, these relationships are more complicated and need further investigation. In our work we illustrate what they look like. The absolute values of $a_d$ in analyzed by us dataset depend on the concentration of organic matter suspended in the water, which is not phytoplankton. Chl$a$-specific NAP absorptions are independent of the SPM concentration, and their values and spectral distributions are determined by particles absorption properties, i.e it's chemical and physical properties (chemical composition, optical properties, sizes, shapes).

According to Ciotti et al. (2002), average chlorophyll-specific light absorption coefficients have the lowest values for microplankton, then nanoplankton, and ultraplankton, and the highest values for the specific light absorption coefficient for picoplankton. These average spectra were determined by size fractionating for the coastal waters of Oregon, the shelf waters of the Bering Sea, and the Bedford Basin (Nova Scotia, Canada). Such an arrangement of average specific absorption coefficients is related to the packing effect of pigments, where in larger particles there may be shading, and therefore less light

absorption. In this work, for the Baltic data, the average chlorophyll-specific absorption coefficients are the lowest for nano-, then ultra- and picoplankton, and the highest for microplankton. Average values of $a_{ph}^{(Chla)}(443)$ for micro, nano, ultra and pico-size classes by Ciotti et al. (2002) were approximately 0.012 m$^2$ mg$^{-1}$, 0.03 m$^2$ mg$^{-1}$, 0.042 m$^2$ mg$^{-1}$ and 0.068 m$^2$ mg$^{-1}$,

respectively. For the data analyzed in this article, the values of $a_{ph}^{(Chla)}(443)$ for the micro, nano, ultra and pico-size classes are characterized by higher values, respectively 0.122 $m^2$ $mg^{-1}$, 0.057 $m^2$ $mg^{-1}$, 0.082 $m^2$ $mg^{-1}$ and 0.112 $m^2$ $mg^{-1}$.

The average determined for the micro-size class is probably overestimated and this applies to all absorption coefficients $a_p$, $a_d$, and $a_{ph}$. This is due to the fact that the Chl$a$_fraction/Chl$a$ or SPM_fraction/SPM predominance condition was met only in 2 cases, the absorption spectra of which differed significantly from each other. Both cases relate to samples collected at the pier in Sopot, one at the end of March (SF04 - Secchi disk was 5.5 m, the bottom was 6 m) and the other at the end of August (SF13 - Secchi disk was 1.5 m, the bottom was at 6 m, there was a large wave, so there was probably mixing
with the bottom particles as well). For the SF04 sample, the concentrations of Chl$a$ and SPM were 0.78 mg $m^{-3}$ and 1.99 g $m^{-3}$, respectively, for the original samples, and the light absorption coefficients of the particles were low. For the micro size class Chl$a$ and SPM were 0.41 mg $m^{-3}$ and 0.92 g $m^{-3}$, respectively. However, for the SF13 sample, Chl$a$ and SPM were 4.05 mg $m^{-3}$ and 15.24 g $m^{-3}$, respectively, for the original samples, and the light absorption coefficients of the particles were an order of magnitude higher than for SF04. For the micro size class, the Chl$a$ and SPM concentrations were 1.87 mg $m^{-3}$ and 8.97 g
$m^{-3}$, respectively.

    As mentioned above, Mohammadpour et al. (2017) studied the optical properties of the SPM size fractions in the littoral waters of Quebec for the size classes 0.2–0.4 μm, 0.4–0.7 μm, 0.7-10 μm and >10 μm. Among other things, mass-specific coefficients of light absorption by various size fractions of suspensions for different regions of the studied water body were presented. Due to the difference in the determination of size fractions, we can only conclude that the tested waters within
Quebec are dominated by mineral suspensions, while the particles from the Baltic Sea waters analyzed in this work are dominated by organic suspensions. Mohammadpour et al. (2017) report spatial differentiation of the contribution of different size fractions of the suspension, which results in a large differentiation of light absorption coefficients. For the 0.2-0.7 μm size classes, this variability ranges from close to zero to > 0.4 $mg^2$ $g^{-1}$ at 400 nm wavelength. Similar differentiation was observed for the fraction >10 μm. Mohammadpour et al. (2017) did not report averaged specific mass factors $a_p^{(SPM)}(\lambda)$. However, their
variability is similar to that presented in the Baltic Sea dataset, except that our spectra are characterized by clear chlorophyll maxima in the green (400-460 nm) and red (660-690 nm) bands.

## 5 Conclusions

    Our research presented in this paper provided important information on the role of particle size and composition in light absorption by size fractions of suspensions in the waters of the Baltic Sea. Particles of different size classes may show
large variability in the absorption properties, with the ranges of variability for small and large particles overlapping. This can cause difficulties in identifying the particle size class on the basis of the light absorption spectrum.

    In this work, for the first time in the history of research on the optical properties of the Baltic Sea waters, we have quantitatively shown different budgets/proportions of light absorption by particles of different size classes based on the analysis of 38 sets of data from both open waters and coastal areas of the Baltic Sea. Despite the complicated relationship between the

influence of particle size and composition on light absorption, we observed that particles with sizes < 5 μm often had a major contribution to $a_p(443)$, $a_d(443)$ and $a_{ph}(443)$, which was more than 60 %. The particle sizes from 5 μm to 20 μm had contribution of $a_p(443)$, $a_d(443)$ and $a_{ph}(443)$ > 19 %, 27 % and 18 %, respectively. On the other hand, particles > 20 μm had contributions in $a_p(443)$, $a_d(443)$ and $a_{ph}(443)$ > 6 %, 4 % and 8 %, respectively.

Measurements of mass-specific and chlorophyll-specific light absorption coefficients of suspended particles, including detritus and phytoplankton, are essential for developing optical inversions to mapping biogeochemical components found in surface waters and to better understand the origin of optical signatures in remote sensing studies. We determined the relationships $a_p$, $a_d$, $a_{ph}$ vs Chl$a$ and SPM for the wavelength of 443 nm. This type of relationship may be useful in local and regional studies of the biogeooptical properties of suspensions. The relationships analyzed in this paper could be used to create/improve a model of light absorption by suspension particles of various sizes in the Baltic Sea. In order to propose such a model, further research is needed, supported in addition by the analysis of particle size distributions (PSD).

So far, the literature has not presented comprehensive analyzes of the variability of light absorption coefficients by different size fractions of particles suspended in seawater. Most studies on fractionation by particle size, mainly phytoplankton, have been carried out by determining Chl$a$ concentrations and then the value of primary production (Decembrini et al., 2014; Deng et al., 2022; Hamasaki et al., 1998; Kormas et al., 2002; Maranon et al., 2001; Cermeno et al., 2005). On the contrary, for all studies of suspended marine solids, size fractionation was carried out mainly to determine the chemical composition of suspended solids (Lam et al., 2017, 2018; Xiang and Lam, 2020; Yigiterhan et al. 2020). In turn, Koestner et al. (2019) investigated the effect of particle size and composition on light scattering in fractionated seawater samples for coastal and estuarine waters in the region of San Diego, California. The relationships of $a_p$, $a_d$, $a_{ph}$ coefficients on Chl$a$ or SPM were shown only for the total suspension, not taking into account the size. However, the relationships presented by other authors were obtained indirectly, but on the basis of, for example, the analysis of diagnostic pigments (DPA) and the size structure (Vidussi et al., 2001; Uitz et al., 2008) or on the basis of HPLC data and the pigment size-class model (Brewin et al. 2010). Brewin et al. (2011) extended the two-factor model of Sathyendranath et al. (2001) and Devred et al. (2006) to a three-component model of light absorption by size fractions. A similar task was undertaken by Hirata et al. (2008) and Aiken et al. (2009). Only studies by Ciotti et al. (2002) showed the actual variability (not modeled) of light absorption coefficients by phytoplankton in size classes for waters of the Bering Sea and coastal waters of Oregon. While Mohammadpour et al. (2017) presented the variability of the light absorption coefficients of all suspended particles in size classes (0.2-0.4 μm, 0.4-0.7 μm, 0.7-10 μm, > 10 μm) in estuarine waters of the Saint Lawrence River and a major SLE tributary, the Saguenay Fjord.

This work is unique not only when it comes to studying the optical properties of the Baltic Sea waters, but also stands out in the international arena of research on the absorption of light by particles in various size fractions. Its main advantage is that it is based on real (not modelled) data. The analyzes presented by us are an introduction to further research, in which HPLC data and the actual particle size distribution should be taken into account in order to formulate an absorption model for particles in different size classes, similarly to Devred et al. (2006, 2011) and Brewin et al. (2010, 2011).

Presented and analyzed in this paper data base of individual particle size fractions in the total SPM, Chl$a$ and related absorption properties for the southern part of the Baltic Sea is relatively small as an effect of time consuming study methods.

However, the conducted research is unique and significant in the study on carbon bulk in Baltic Sea. Particle size determines the proportions of the organic carbon suspended in the water column and deposited to the sediments. The main source of POM in sea sediments are particles smaller than 20 μm (Moynihan et al., 2016). Our research shows that in the surface layer, the POM/SPM ratio is the highest for particles < 5 um (for ultra particles about 72%, for pico particles about 63%). Nano particles transfer about 59% of POM in SPM, while micro particles transfer 53%. These studies show that the process of deposition of

organic particles in Baltic Sea sediments should be ongoing in a similar way as in other coastal areas. Baltic Sea, however, is characterized by a permanent pycnocline (Kowalczuk et al., 2015; Sagan, 2008). Portion of the sinking organic particles never reaches the bottom sediments but is trapped by the halocline and mineralized there. Our research can be helpful with understanding of POM transfer to the sediments of Baltic Sea and calculating its pace and bulk.

Our research indicates the need for further investigation of this topic, also for the remote estimation of the size

structure of phytoplankton populations, the so-called PFT. In case of methods based on spectral characteristics of phytoplankton (e.g. light absorption coefficients) and the dominant particle sizes, many computational steps are necessary. Based on remote sensing reflectance, the inherent optical properties (e.g. total light absorption and backscattering coefficients) of the water can be calculated. Then, in several steps, effective estimation of for example normalized light absorption coefficient of phytoplankton spectrum is possible. On this basis, the characteristics of the phytoplankton size (e.g. size factor,

which determines relative proportions between 'small' and 'large phytoplankton cells in studied population) can be determined.

**Data availability:**

The *in situ* light absorption data (absorption by particles, detritus, phytoplankton and CDOM) and biogeochemical data (concentration of Chl$a$ and SPM) can be acquired for scientific research purposes upon request from Justyna Meler

**Authors contribution:**

Conceptualization, J.M.; methodology, J.M. and M.Z.; validation, J.M, M.Z. and D.L.; formal analysis, J.M.; investigation, J.M.; resources, J.M, D.L., M.Z., data curation, J.M., M.Z,; writing—original draft preparation, J.M.; writing—review and editing, M.Z. and D.L.; visualization, J.M.; supervision, M.Z., All authors have read and agreed to the published version of the manuscript.

**Conflicts of Interest:**

The authors declare that they have no conflict of interest.

**Acknowledgments:**

This research was carried out as part of the project funded by the National Science Centre, Poland, entitled 'Investigating the variability of the spectra of light absorption coefficients by various size fractions of suspended matter occurring in the southern area of the Baltic Sea' (contract No. 2020/04/X/ST10/00335) (awarded to Justyna Meler). We are grateful to the Institute of

Oceanology Polish Academy of Sciences in Sopot, represented by Sławomir B. Woźniak, Joanna Stoń-Egiert and Karolina Borzycka for their help in collecting the empirical material.

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
