# Peer review of "Variability of light absorption coefficients by different size fractions of suspensions in the southern Baltic Sea"

_Biogeosciences, 2022_

## Author Comment (AC1)

Reviewer's comment:
*General comments:*

*Meler et al. present a descriptive study documenting measurements of particulate, phytoplankton, and detrital absorption coefficients obtained from size-fractionated water samples collected from nearshore and offshore waters of the Baltic Sea. The water samples comprise a significant range in biogeochemical properties based on SPM and Chla observations, although the sampling locations were confined to a southern sub-region of the Baltic sea. The dataset contains 38 elements. The results do not indicate significant differences between size fractionated samples in terms of absorption properties. Further, the results indicate differences in mass-specific (but not Chla-specific) light absorption coefficients between larger (micro) versus smaller (nano and pico) organic and inorganic particles.*

Author's response:
We thank the Rewiever for insightful critical comments on our work.

*Although the general topic could potentially be compliant with the journal's scope, the manuscript does not satisfy the journal's criteria to merit publication, as follows:*

*Scientific Significance: The study is not sufficiently comprehensive based on insufficient number of observations (n=38) and small spatial extent of the sampling area, compared with the variety of oceanic conditions, physical forcings, biological conditions, and the terrestrial, riverine, and anthropogenic inputs to the Baltic Sea. The key-finding (that Chla-specific absorption properties of size-fractionated samples are not significantly different from each other within the authors' n=38 dataset from southern Baltic Sea waters near Poland) would be more compelling if the study was more expansive, or if the authors could better establish the significance of their null results. The study may also not be generalizable outside of the Baltic Sea, and because the observations span a small geographical sub-region of the Baltic sea (the southern waters around Poland) the results may also not be representative of the optical properties elsewhere in the Baltic Sea;*

Author's response:
The research results presented in the manuscript include 38 observations collected during 1 year of research conducted during the implementation of a small scientific project. The implementation of 1 measuring station was very time-consuming. Water intake, size fractionation and then filtration of the original water samples and fractions took from 4 to 7 hours, depending on the "purity" of the water. The data was collected during 3 several-day cruises on the Baltic Sea, covering waters with different optical properties, and periodically conducted research at Sopot Pier (monitoring of temporal variability). The research was focused mainly on the Gulf of Gdańsk, where the ranges of variability of optically active components are representative for the southern part of the Baltic Sea, which was shown in earlier works by Meler et al. (2016a and b, 2017). However, 1/3 of the observations during cruises also included stations outside the Gulf of Gdańsk, i.e. open and coastal waters without direct influence of river waters. In the Gulf of Gdańsk, we conducted research at stations located in the plume of the Vistula River (the main, large river that has a large impact on optically active components in the waters of the Gulf) and at stations distant from the mouth of the Vistula River. Based on literature data on other Baltic regions, in particular off the coasts of Sweden, Finland and Latvia, the results presented in our manuscript are rather not representative of these areas, however, knowledge about the diversity of the Baltic waters in the aspects we study is still limited.

The presented results may not be of global interest, however, the Baltic Sea is an important element of the ecosystem for 11 countries of the Baltic Region under the Monitoring and Assessment Strategy of the Helsinki Commission (HELCOM, 2013). The HELCOM strategy is intended to provide assessment and monitoring of data that can be used both for international assessment by HELCOM and for monitoring at the national level. The strategy is designed to ensure both data production and dissemination of information by contracting parties of EU Member States and meeting the requirements of several EU Directives such as the Marine Strategy Framework Directive (MSFD), the Water Framework Directive (WFD), Habitats and Birds Directives, EU Strategy for the Baltic Sea Region (EUSBSR) and EU Integrated Maritime Policy (HELCOM, 2013). First of all, the Strategy aims to support ecosystem-based maritime spatial planning (MSP) in the Baltic Sea by enabling high-quality spatial data and assessment tools for MSP purposes. For the purpose of regional assessment, HELCOM divides the Baltic Sea into different waters. These basins have been described in the document "HELCOM Sub-areas of the Baltic Sea" (Annex 4; HELCOM, 2013), according to separate hierarchical division levels, depending on management needs.

Author's response:
In the new version of the manuscript, we have modified selected parts of the text. We have supplemented the Introduction section with the above information.

Reviewer's comment:
*Presentation Quality: Comprehension of the manuscript is inhibited by low presentation quality. In particular, the authors' combination of the Results and Discussion materials into a single section significantly detracts from the presentation of each, and at times made comprehension of the manuscript difficult, or resulted in ambiguity in elements of the methods or results. I suggest that the authors separate the results and discussion in order to add clarify.*

Author's response:
We agree with the Reviewer that the first draft of the manuscript may have been difficult to read. The manuscript was reorganized following the Reviewer suggestion.

Reviewer's comment:
***Additional (general) comments:***

*The authors do not adequately demonstrate the other dimensions of variability in their dataset, e.g., due to seasonal factors, site-specific differences like onshore vs offshore, biomass, or total particle content. One way that the authors could have helped with this would be to color the markers in the scatter plots to indicate other parameters, e.g., by seasons or by whether the site was nearshore or offshore.*

Author's response:
As suggested by the Reviewer, in Figures 2-8, we marked the season and the sampling area with colors and various markers. In the case of cruise data, the division is as follows: February-winter, April-spring, September-autumn, and spatial division into samples from the Gulf of Gdańsk and open and coastal waters. In the case of data from Sopot Pier, the data is presented as a separate group covering all 4 seasons: winter (December 21 - March 20), spring (March 21 - June 22), summer (June 23 - September 20) and autumn (September 21 - December 20th).

We supplemented the Material and Methods section with a description of the sampling area, where we presented the seasonal cycle of biological activity in the Baltic Sea, as well as a division into regions and a description of hydrological conditions.

The principal factor affecting the variability of the inherent optical properties of Baltic waters in the euphotic zone is the seasonal cycle of biological activity (Sagan, 1991; Olszewski et al., 1992; Kowalczuk et al., 1999, 2005; Meler et al., 2016 a,b). This cycle is governed by physical, biological and chemical processes, which cause the biomass and species composition to vary with time. As a consequence there are three main phytoplankton blooms: a spring bloom of cryophilous diatoms, which then transforms into a bloom of dinoflagellates; a summer bloom of cyanobacteria; and an autumn bloom of thermophilous diatoms (Thamm et al. 2004, Wasmund et al. 2001, Witek and Pliński 1998). The spring blooms can take place from early March to May, the summer ones in July/ August and the autumn ones from September to October (Wasmund et al. 1996, Thamm et al. 2004, Wasmund and Uhlig 2003). In winter, biological activity is minimal. The maximum runoff of river waters occurs at the turn of April and May, and it often coincides with the spring bloom of phytoplankton initiated by an increase in air and water temperature and more sunlight. River waters carry large amounts of organic dissolved substances (DOM) and nutrients that enhance phytoplankton blooms. The increased amount of phytoplankton in the surface water layer reduces the transparency of the water. In summer there are periodic floods following very heavy rainfall which together with strong winds can effect in upwelling, which causes cooler water to rise up from the deep layers of the sea. Such events affect the optical properties of waters in the coastal zone and Gulf of Gdańsk - see Olszewski et al. (1992), Kowalczuk (1999) and Sagan (2008).

Locations of sampling stations in our study were selected to obtain the greatest possible diversity of waters in optical terms. The research was carried out mainly in the Gulf of Gdańsk, but also in open waters and coastal waters outside the Bay of Gdańsk, as the weather permits.

The analyzed data set was divided due to the sampling area: the Gulf of Gdańsk (GG), and extracted from the GG - Sopot Pier (SF) (which shows time variability over 10 months and is the only one that takes into account the summer season), open and coastal waters (OCW) In addition, the data were divided due to the season of sampling. In the case of cruise data, the division is as follows: February - winter, April- spring, September - autumn. In the case of data from Sopot Pier, the data are presented as separate group covering all 4 seasons: winter (December 21 - March 20), spring (March 21 - June 22), summer (June 23 - September 20) and autumn (September 21 - December 20). For the 14 samples (collected during cruises in February and April), it was not possible to separate the ultra and picoplankton fractions, because the amount of suspension of 2-5 µm clogged the membrane filters.

Reviewer's comments:
*The authors did not identify differences in Chla-specific optical properties between size fractionated samples. I'd suggest that the authors investigate or discuss what other factors (e.g., distance from shore, biomass, wind-driven mixing, contribution of inorganic particles) may have been associated with the variability in observed Chla-specific absorption properties within size fractions.*

Author's response:

The average Chla-specific absorptions (as well as mass-specific absorptions) presented by us were determined only for samples in which a given fraction was dominant, and not for all samples for a given fraction. This was to try to show the average absorption spectra for a given particle size class, similar to what Ciotti et al. (2002) for waters that were optically less complex than the Baltic Sea we studied. As suggested by the reviewer, we divided the data set into seasons and sampling areas.

As suggested by the Reviewer, we have shown in Figures 2-3 and 5-8, the division into seasons and sampling areas.

We have added Table 2 showing the proportions of SPM and Chla in size classes (micro, nano, ultra, pico, or ultra+pico) in total SPM and Chla, for all data and divided by regions (Gulf of Gdańsk, Sopot Pier and open and coastal waters).

We have also extended the POM/SPM description to seasonal and spatial division.

Descriptions to corrected figures 5-7 have been extended.

An extended description for Figure 8 is presented in text. Table 3, presenting the contributions of particles from different size classes to the total light absorption by all particles, detritus and phytoplankton, has been modified by sampling regions.

New Figure 2

[revised manuscript text omitted]

New Figure 3

In Figure 3, for all data, it can be seen that the winter season, regardless of the place where POM/SPM samples were taken, assumes the lowest values from SPM. No trends were observed between the remaining seasons and sampling sites. In the case of micro particles, most of the GG samples in autumn are dominated by inorganic particles (POM/SPM < 25%). For nanoparticles, in the winter, inorganic matter dominated, and in the autumn, organic matter dominated. For ultra particles, no seasonal and spatial dependencies of POM/SPM vs SPM are visible. On the other hand, for pico particles, we observed that POM/SPM increases with the increase in SPM: in winter POM/SPM had the lowest values, then in spring and autumn it was on average 65% and the highest values reached in summer on average 80%.

[Figure]

**Figure 3: Relationships between the POM/SPM ratio and the SPM concentration for the original water samples and the size classes: micro, nano, ultra and pico. Mean values ± standard deviation are shown in the graph. Markers shapes and colours distinguish the season and sampling site (GG - Gulf of Gdańsk, SF - Sopot Pier, OCW - open and coastal waters).**

New Figure 5

[revised manuscript text omitted]

As suggested by the Reviewer, we have divided the Results and Discussion sections into two separate sections. In its current form, the Discussion section is as follows.

[revised manuscript text omitted]

Reviewer's comments:
*Comparing the overlap in mean +/- std between data points is most useful when uncertainties due to environmental or methodological variability are well described (uncertain measurements of moderately dissimilar parameters can easily overlap). The authors do not convey uncertainty in their absorption, Chla, or SPM measurements, which would help to identify the extent to which overlap in absorption properties is or is not meaningful.*

Author's response:
In the case of the analyzed data set, the precision of the measurements of the light absorption coefficients and the concentration of chlorophyll a was not checked, because no duplicate samples were made. Checking the precision of the measurements of these parameters previously performed on a different dataset yielded the following results.
The precision of the measurement of light absorption coefficients using the IS method for 3 different filters from the same station was 4.96% +/- 2.91%. When measuring the concentration of chlorophyll a for duplicate seawater samples, the measurement precision was 5.3% +/- 1.5%. In the case of SPM, according to the methodology, 3 subsamples are always taken and the measurement precision for 95% of the triplets was below 15%, and for all cases the average was 5.83% +/- 4.40%.

In the Materials and methods section, appropriate descriptions of measurement precision have been added.

Reviewer's comments:
***Minor (Specific) Comments***:

*Table 1: Is the section "Nano+ultra particles (2-20um)" intended to be Pico + nano particles (based on the sampling difficulty of the first 14 samples; L200-202)?*

Author's response:
Nano+ultra particles refers to the classic division into size classes according to Sieburth et al. (1978), where particles with a size of 2-5 μm were still treated as nanoplankton and did not constitute a separate size class. In the case of pico + ultra particles, due to too much of these particles, the membrane filters were clogged and had to be replaced too often, and it was not possible to filter enough water volumes for filtration to obtain SPM and Chl*a*, due to limited funds and time.

Reviewer's comments:
*Lines 303-312 and figures 5-7: I'd suggest that log scale R2 values are reported as well. These datasets are mostly log-normally distributed in both axes, and R2 calculated on the linear axes is strongly influenced by the points in the upper-right corner of the plot. For example, consider the high R2 despite low association of points in Fig 6 panel G.*

Author's response:
Figures 5-7 show the dependence of the light absorption coefficients of all particles, detritus and phytoplankton at 443 nm on the Chl*a* and SPM concentrations on the log-log scale. The presented approximations are a power function $y=A*y^B$ and the coefficients $R^2$ correspond to these approximations.

We added this information in section 3.3.

---

## Author Comment (AC2)

Reviewer's comment:
*The manuscript by Meler et al. investigated the size-fractionated absorption spectra of particles, phytoplankton, and non-algae particles (NAP) in the southern Baltic Sea. They also conducted the measurement of total and size-fractionated suspended particulate matter (SPM) and Chlorophyll (Chl) a concentrations and then examined the relationships between the absorption coefficients, SMP, and Chl a concentrations for each size fraction. They found that the SPM-specific absorption coefficients are a useful parameter to distinguish between large and small plus medium particle fractions. The data presented in this study is informative. However, this manuscript requires considerable alteration along the lines I have suggested below.*

Author's response:
We thank the Rewiever for insightful critical comments on our work.

We modified selected fragments of the text and figures, in accordance with most of the Reviewer's suggestions.

Reviewer's comment:
**Major comments**
*The description of total and size-fractionated Chl a-specific NAP absorption needs more detail. It is possible to understand the meaning for calculating the absorption coefficients of particles ($a_p$) and phytoplankton ($a_{ph}$) normalized by Chl a and SPM concentrations to see the contribution of each size component to the spectral shape and magnitude. However, I am not sure the significance of the Chl a-specific NAP absorption spectra and coefficient at 443 nm as shown in Figures 6a – e, and 9c, 10c.*

Author's response:
Figure 6 shows $a_d(443)$ vs Chl$a$ and vs SPM coefficients, while Figure 9c and 10c show chlorophyll-specific $a_d(443)$. In general, if we can express $a_p$ and $a_{ph}$ in the form of a Chl$a$-dependent function, then so can $a_d$, since $a_p = a_d + a_{ph}$. For clean ocean waters (the so-called Case1) Bricaud et al. (1998) presented Chl$a$-specific NAP absorption $a_d(\text{Chl}a)$ can be determined from the difference of $a_p(\text{Chl}a) – a_{ph}(\text{Chl}a)$ (Woźniak & Dera, 2007). In the case of the optically complex Baltic Sea, these relationships are not so simple, and we just wanted to illustrate what they look like. For the analyzed data set, on average, 52% of suspended matter was organic matter of both autogenic and allogeneic origin, and the contribution of inorganic matter to light absorption is not significant (Woźniak and Dera, 2007). Therefore, the absolute values of ad in the analyzed set depend on the concentration of organic matter suspended in the water, which is not phytoplankton. Chl$a$-specific NAP absorptions are independent of the SPM concentration, and their values and spectral distributions are determined by the absorption properties of the suspension particles themselves, i.e. they depend on the chemical and physical properties of the material they are made of (chemical composition, optical properties, sizes, shapes).

Earlier, we heard the opinion that the concept of $a_{NAP}$, or "particles other than algae", is too empirical, because it consists of an unknown admixture of mineral and organic detritus. The very different refractive indices of mineral and organic matter make it impossible to interpret changes in $a_{NAP}$ in any quantitative way. I mass-specific $a_{NAP}$ based on SPM are of no value without the partitioning of SPM into PIM and POM (Duarte et al. 1998, Richter and Stavn,

2014). On the other hand, chlorophyll-specific $a_{NAP}$ may be of some value. Therefore, in our work we decided to show both approaches.

Reviewer's comment:
*I agree with the author's assertion that the data obtained by this study could improve the model to retrieve the inherent optical properties (IOPs) in the Baltic Sea (Lines 464 – 465). However, it is not clear that which of the results or relationships examined in this study would contribute to the improvement of the IOPs models and how to expand the results into the models for estimating the size parameters. Given that many cases have already been reported in the literature (as cited by the authors themselves in the Conclusion section), it would be advisable to explain specific information on the improvement of IOP models.*

Author's response:.
Our research results are preliminary and very limited.
The analyzes presented by us are an introduction to further research, in which HPLC data and the actual particle size distribution should be taken into account in order to formulate an absorption model for particles in different size classes, similarly to Devred et al. (2006, 2011) and Brewin et al. (2010, 2011).

Our pilot studies on the study of the contribution of individual particle size fractions in the total SPM, Chla and related absorption properties for the southern part of the Baltic Sea indicate the need to develop this topic, especially for the remote estimation of the size structure of phytoplankton populations, the so-called PFT.

These sentences have been included in Conclusions.

Reviewer's comments:
*The large part of the sentences in the Introduction reviews the previous literatures. Therefore, it seems to me that it is hard from reading the Introduction to understand why this study is needed. To better organized the introduction and objectives, I would encourage the authors to rewrite the section. Similarly, abstract and most parts of results and discussion sections, especially 3.2, 3.3, and 3.4, are not well organized. It is descriptive and is like a data report, making difficult to follow what is the new findings described in this study. However, I believe that the authors can elaborate.*

Reviewer's comments:
*Figure 2a showed the results of size-fractionated "SPM" in each sampling station. A more appropriate legend would be required for Figure 2a to better reflect the investigation of SPM.*

Author's response:
We agree with the opinion of the Reviewer. Sections Introduction, 3.2, 3.3, 3.4 have been modified. Figures and tables have been modified. The descriptions were extended with the seasonal and spatial division of the analyzed data set.

In the Introduction section, we have completed the goals of our research.

Various approaches to identify the size structure of phytoplankton populations from satellite data are detailed in the IOCCG report (2014), which describes the possibilities of developing algorithms for remote determination of the contribution of various functional types of phytoplankton (PFT) included in the total population in the waters under study. The PFT concept is used in the study of a number of ecological and biogeochemical problems, especially in model studies. A specific functional type may represent a group of different species related to each other due to certain distinguished features. This approach is of growing interest as it allows for a more thorough study of the role of phytoplankton in global sea and ocean cycles involving the circulation of major chemical elements such as carbon, nitrogen, sulfur and iron, as well as photosynthesis and primary production.

The above mentioned methods of estimating the contribution of phytoplankton size classes do not work for the Baltic Sea, which is a reservoir classified as optically complex (for the DPA method, the results are presented in Meler et al., 2020).

(...)

For this purpose, *in situ* studies should be carried out, which would allow to directly determine the light absorption coefficients not only by phytoplankton, but also by all particles and detritus in various size fractions. The research described in this paper is in line with the objectives and guidelines of the Monitoring and Evaluation Strategy of the Helsinki Commission (HELCOM), which aims to ensure the evaluation and monitoring of data that can be used by HELCOM, both for international and national monitoring. The strategy is designed to ensure data production and dissemination by contracting parties of EU Member States. These countries are obliged to comply with several EU directives, such as the Marine Strategy Framework Directive (MSFD), the Water Framework Directive (WFD), the Habitats and Birds Directives, the EU Strategy for the Baltic Sea Region (EUSBSR) and the EU Integrated Maritime Policy ( HELCOM, 2013). First of all, the Strategy aims to support ecosystem-based maritime spatial planning (MSP) in the Baltic Sea based on ecosystem. It is done by enabling high-quality spatial data and assessment tools for MSP purposes.

Our research may be useful in examining whether the use of SPM and Chla data from MERIS or other optical sensors installed on satellites (e.g. OLCI - Ocean and Land Color Instrument) can be used as "high-quality spatial data" and as a HELCOM regional assessment tool.

We have shown in Figures 2-3 and 5-8, the division into seasons and sampling areas. And the relevant comments are placed in the text of manuscript.

New Figure 2

[revised manuscript text omitted]

Reviewer's comments:
*Although average Chl a-specific absorption coefficients of phytoplankton generally decrease with increasing cell size because of self-shading, the authors showed the opposite trends as compared with previous work of Ciotti et al. (2002). Therefore, I feel that the package effect (as mentioned by the authors themselves in Line 410) may be open to further discussion.*

Author's response:
Figure 9 and 10 shows the average spectra of specific light absorption coefficients by all particles, detritus and phytoplankton for given fractions, determined for cases where a given size fraction was dominant (it is not an average for all measured coefficients for a given fraction). In the analyses, we used Chl$a$ determined by spectrophotometry, not by HPLC, so we do not know the share of individual pigment groups, so we are unable to determine the packing effect. Of course, taking into account that the largest share of micro particles was recorded for the SF13 station during phytoplankton bloom (large algae gathered at the beach in Sopot), we can assume that the packing effect occurred and was significant.

Reviewer's comments:
***Minor Comments:***

*Names of observed stations are missing in Figures 1, which make it difficult to refer to Figures 2 and 8 and SF04 and SF13 in Lines 417 – 427. The information will help readers understand the results more easily.*

Author's response:
SF01-SF16 refer to measurements on Sopot Pier and constitute a separate group showing temporal variability. We have improved the descriptions in the text. Figure 1 has been modified, station names have been added.

New Figure 1

[Figure]

**Figure 1: Location of measurement stations in the Baltic Sea.**

Reviewer's comments:
*I would suggest that the results of Figure 4, 9, or 10 be presented in a different way; for example, a box plot at satellite ocean colour bands with average spectra could be used. I think that this make it easier for the readers to understand the importance of them. For example, please refer to Brunelle et al. (2012, doi: 10.1029/2011JC007345).*

Author's response:
We have presented Figure 4 as suggested by the Reviewer, as far as possible. However, Figures 9 and 10 in boxed form for selected wavelengths do not work in our case. The variability ranges of individual means and standard deviations for size classes overlap, obscuring the picture. Therefore, in the existing drawings, we have bolded the average values for better visibility, and in the Table 4 we have compiled numerical values for selected wavelengths, corresponding to the ranges observed by satellite sensors such as Seawifs or OLCI.

New Figure 4

[revised manuscript text omitted]

---

## Author Response (AR2)

Dear Editor,

We thank the Reviewers for their time devoted to our manuscript "Variation of light absorption coefficients by fractions of various sizes of suspensions in the southern Baltic Sea" by Justyna Meler, Dagmara Litwicka and Monika Zabłocka. We are glad that we were able to satisfactorily respond to most of the comments of both Reviewers.

We followed the directions of Reviewer#1. We modified the abstract and made some changes to the Introduction and Conclusions to better emphasize the importance of our research.

Please find our detailed response for Review#1.

**Reviewer#1:**

Reviewer's comment:

I would like to thank the authors for their sincere efforts to address my concerns. However, I wonder if it can be any more generalized conclusions based on this study. I think that the readership of Biogeosciences would expect broader implications. As I mentioned in the previous review, the manuscript is descriptive, making it difficult to follow what are the new findings described in this study. Especially, the abstract is still descriptive and should be modified again. Furthermore, although I agree with the significance of the HELCOM monitoring program in the Baltic Sea, it seems to me that the pilot studies and/or an introduction to further research (which was mentioned in Author's response to my major comment No.2) are not satisfactory for the journal's criteria to merit publication in Biogeosciences.

Author's response:

We thank the Reviewer for constructive comments.

We modified abstract to better highlight the main findings of this work. Our research mainly concerned the variability of light absorption coefficients depending on the size fraction of particles suspended in seawater. However, we also showed the contributions of individual fractions to the total concentrations of SPM and Chl*a*. Knowledge of particle sizes and their biogeochemical and optical properties in given size fractions is the basic knowledge that leads further to the carbon study in the Baltic Sea. The size of the particles determines the amount of organic matter that will be transported deep into the sediments. For example, pico and nanoplankton in clean ocean waters is the main carrier of POM deposited in the sea bottom. Therefore, particle size plays a very important role in the efficiency of a biological pump. This fact alone is an important justification for undertaking such research for the waters of the Baltic Sea, which is characterized by different biogeochemical and optical properties than ocean waters.

Presented and analyzed in this paper data base of individual particle size fractions in the total SPM, Chla and related absorption properties for the southern part of the Baltic Sea is relatively small as an effect of time consuming study methods. However, the conducted research is unique and significant in the study on carbon bulk in Baltic Sea. Particle size

determines the proportions of the organic carbon suspended in the water column and deposited to the sediments. The main source of POM in sea sediments are particles smaller than 20 μm (Moynihan et al., 2016). Our research shows that in the surface layer, the POM/SPM ratio is the highest for particles < 5 um (for ultra particles about 72%, for pico particles about 63%). Nano particles transfer about 59% of POM in SPM, while micro particles transfer 53%. These studies show that the process of deposition of organic particles in Baltic Sea sediments should be ongoing in a similar way as in other coastal areas. Baltic Sea, however, is characterized by a permanent pycnocline (Kowalczuk et al., 2015; Sagan, 2008). Portion of the sinking organic particles never reaches the bottom sediments but is trapped by the halocline and mineralized there. Our research can be helpful with understanding of POM transfer to the sediments of Baltic Sea and calculating its pace and bulk.

Reviewer's comment:

In addition, I am a bit confused about the sentences explaining Phytoplankton Functional Types (PFTs). The manuscript focuses on the size components of absorption spectra of particles, NAP, and phytoplankton as well as both Chl a and SPM concentrations so that this study could contribute to the further understanding of Phytoplankton Size Classes (PSCs). In this sense, it is difficult to follow the logic in the Introduction and the last sentences in the Discussion section explaining PFTs. To better establish consistency, these sentences should be modified.

Author's response:

Indeed, the manuscript does not focus on PFT. Our research is general and shows the variation in light absorption coefficients for all particles, not just phytoplankton. Our goal was to show that part of our research can be used to determine the size structure of phytoplankton in the Baltic Sea.

In the Introduction we have added appropriate supplementary sentences in lines 35-45 and 120-123, and in the Conclusion in lines 670-690.